# Towards non-blind optical tweezing by finding 3D refractive index changes through off-focus interferometric tracking

Benjamin Landenberger[1,2], Yatish[1,3,4] & Alexander Rohrbach [1,2,3 ✉]

In modern 3D microscopy, holding and orienting arbitrary biological objects with optical forces instead of using coverslips and gel cylinders is still a vision. Although optical trapping forces are strong enough and related photodamage is acceptable, the precise (re-) orientation of large specimen with multiple optical traps is difficult, since they grab blindly at the object and often slip off. Here, we present an approach to localize and track regions with increased refractive index using several holographic optical traps with a single camera in an off-focus position. We estimate the 3D grabbing positions around several trapping foci in parallel through analysis of the beam deformations, which are continuously measured by defocused camera images of cellular structures inside cell clusters. Although non-blind optical trapping is still a vision, this is an important step towards fully computer-controlled orientation and feature-optimized laser scanning of sub-mm sized biological specimen for future 3D light microscopy.

[1] Laboratory for Bio- and Nano-Photonics, Department of Microsystems Engineering-IMTEK, University of Freiburg, 79110 Freiburg, Germany. [2] BIOSS Centre for Biological Signalling Studies, University of Freiburg, Freiburg, Germany. [3] CIBSS - Centre for Integrative Biological Signalling Studies, Freiburg, Germany. [4] Spemann Graduate School of Biology and Medicine (SGBM), University of Freiburg, Freiburg, Germany. ✉email: rohrbach@imtek.uni-freiburg.de

When we buy fruits in a super-market, we often grab, rotate and squeeze the product with our hands to investigate it from all sides. While this process is relatively easy in a macroscopic world, a precise control of object orientation is still a big challenge in the microscopic world. Especially in 3D microscopy, this so far non-existing feature would be very beneficial for a visual object inspection.

The behavior of cells on a glass coverslip is often significantly different to cells embedded in a natural matrix of adjacent cells. Therefore the preparation, handling, and the investigation of cell communication and responses to stimuli require more advanced manipulation for imaging methods, such as e.g. two-photon microscopy[1] or light-sheet microscopy[2]. Without toxic clearing techniques[3], cell clusters, small plants or embryos with sizes from tens of μm to few mm are often so thick that illumination light and/or fluorescence light is absorbed or scattered so strongly[4] that observation from different directions is often the only way out[5]. Embedding the specimen in soft gels (e.g. agarose) prevents unwanted motion or diffusion in liquid environments. One-axis rotatable gel cylinders can make alignment procedures complicated, hinder object growth and make multi-object investigations nearly impossible.

Repositioning and reorientation of objects with electromagnets can be achieved by adding micro-magnets to millimeter-sized specimen either directly or inside rotatable gel spheres embedding the biological object[6]. Optical forces, adequately distributed across the specimen as sketched in Fig. 1a, can work even without adding handles. On the scale of a few cells multiple, typically holographic optical tweezers[7] are the most flexible tool to reorient and rotate objects without mechanical contact like invisible fingers[8,9], However, reaching several local potential minima is not possible with multiple optical traps and most objects, which are hardly deformable - leading to blind or frustrated trapping.

Several approaches have been pursued to trap, hold and orient biological objects with optical forces to improve imaging. Single cells or bacteria have been rotated by multiple-point holographic optical traps during brightfield imaging[10,11], which can also reorient single bacteria during conventional fluorescence[12] or super-resolution fluorescence[13] imaging. Based on a balance of scattering forces, objects were shifted by four beams to allow multiview imaging[14] or by two fiber-emitted counter-propagating beams, where additional cell rotation was achieved by rotating the asymmetric beams[15]. As an alternative to holographic trapping, time-multiplexed point traps have allowed to hold and manipulate deforming bacteria[16] or single microtubules[17] in combination with fast super-resolution coherent imaging. Large specimen, with tens of micrometers in diameter, have been trapped and oriented by using low focused, counter-propagating laser beams enabling a large working range in combination with acoustic forces and conventional imaging[18] or with light-sheet fluorescence imaging[19].

However, in all these approaches, highly precise 3D control in position and orientation has not been possible, mainly because of a missing feedback, also called force clamp. This was partly achieved in early approaches with elongated particles (diatoms) with optimized video control[20], for a position clamp and single-axis rotation[21] or counter-propagating beams[22]. Using time-multiplexed optical traps and electro-optical tunable lenses, Tanaka et al.[23] realized optical multiple-force clamps for 3D rotational control of diatoms and their fragments. Surface relief imaging approaches used dynamic optical traps to scan particles across a structured surface requiring advanced feedback mechanisms to control the position of either small spheres[24] or complex-shaped styluses using force feedback[25]. An advanced method to rotate small objects performs a 3D tomographic scan to reconstruct the 3D refractive index (RI) distribution of cells of arbitrary shape[26]. From this, a 3D coherent light distribution is calculated and generated holographically in real time, which exerts optical torques in different directions. However, because of the intensity

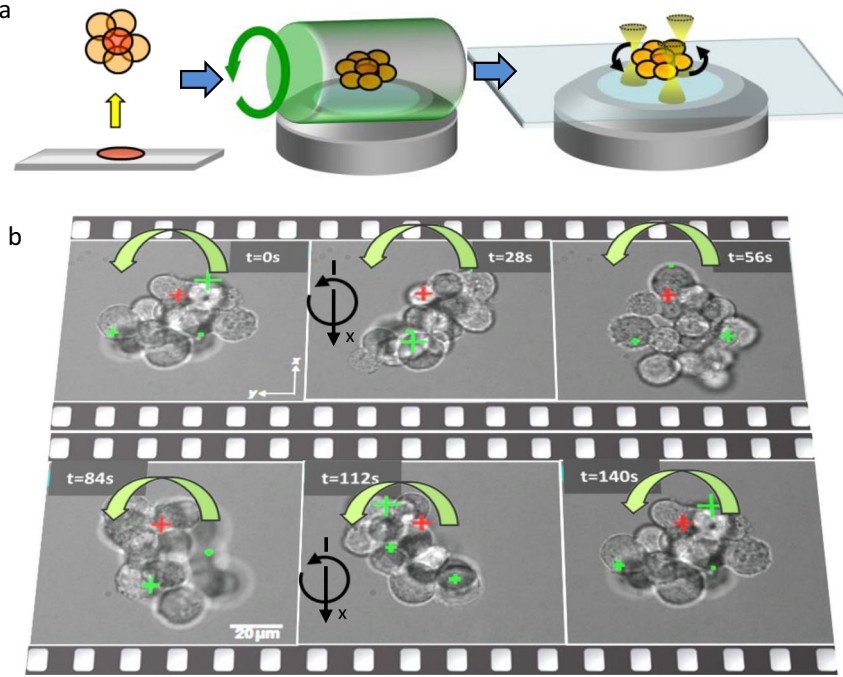

**Fig. 1 Positioning and orientation of biological specimen. a** Cell cluster without coverslip, in a rotating gel cylinder and contact-less in multiple optical tweezers. **b** Time-lapse from a 70 μm large cell cluster rotated around the *x*-axis (parallel to the image plane) by three dynamic, but blind optical traps (see supplementary movie 3). The xy position of the rotation center is marked by a red cross, the changing centers of the optical traps by green crosses (with marker sizes proportional to the axial position).

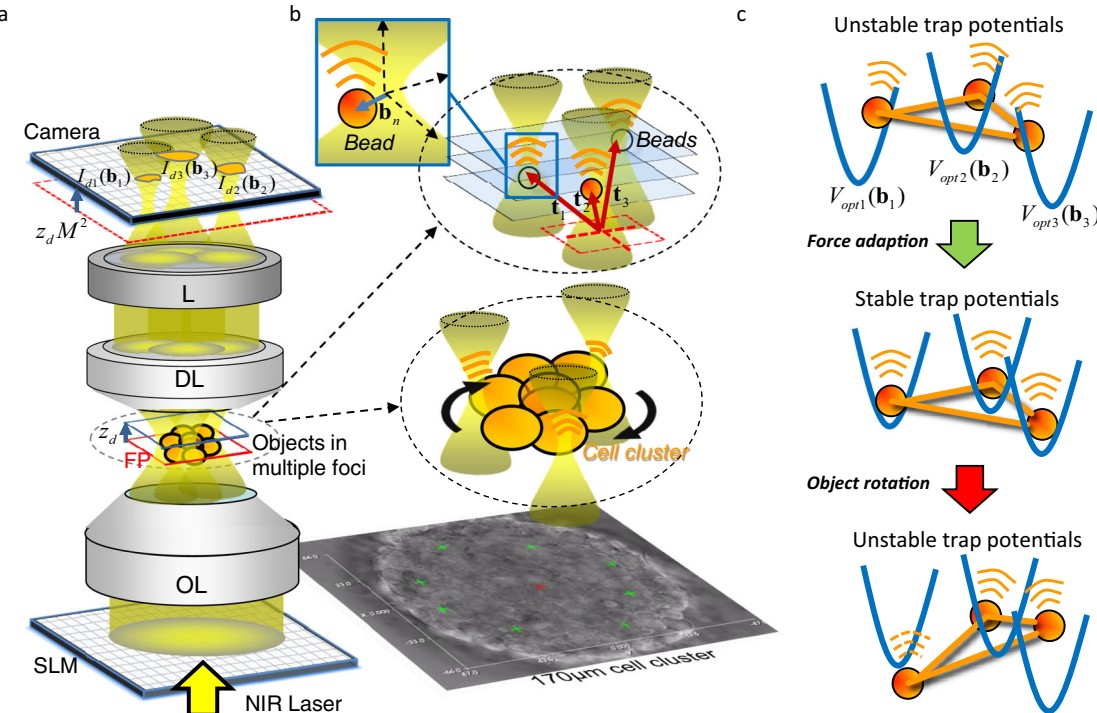

**Fig. 2 Working principles. a** Setup sketch and principle of OFI-tracking. A spatial light modulator (SLM), illuminated by a near-infrared (NIR) laser beam generates several optical traps around the focal plane (FP, in red), which are imaged telecentrically with two lenses (DL and L) onto a CCD camera. **b** Objects within the FP—a cell cluster or beads—scatter the focused light resulting in a redistribution of intensities $I_{dn}(\mathbf{b}_n)$ at the camera, which encode the object positions $\mathbf{b}_n$ relative to the beam centers. The vectors $t_n$ indicate the trap positions relative to the center of the FP. Bottom: brightfield image of a 170 μm large cell cluster held by eight optical traps (marked by eight green crosses. **c** Blind and non-blind optical trapping. An extended stiff object is (blindly) trapped by three unstable optical traps. After adaptation of the focus positions $t_n$ according to the OFI-signals, trapping becomes stable (non-blind trapping). Object rotation by shifting the laser foci, makes trapping unstable again (blind or frustrated trapping).

gradients, which are manifold weaker than those of point traps, only small objects such as a few cells can be rotated.

The driving optical forces come to their limits for increased gravitation and friction forces—counteracting the rotation of cell clusters or small embryos. Figure 1 demonstrates what is currently the limit of blind (holographic) optical trapping. Object rotation can be achieved with a single or a few cells (Supplementary Movie 1 and 2), but is in principle also possible—although more difficult—for larger cell clusters as demonstrated in Fig. 1b and Supplementary Movie 3. However, without knowing the optimal grabbing positions, we translate several optical point traps across a spherical surface to rotate a biological specimen out of the focal plane, i.e., using the weak axial gradient forces, and to hope that the particle will drift-diffuse sufficiently deep into the optical trapping potential (Fig. 2c).

Thus, non-blind optical trapping means that an efficient trapping position within the RI distribution is found by any readout signal to enable stable object displacements and small rotation steps. Non-blind optical trapping with point tweezers is still a vision and might be achieved within the next 5–10 years. It is of great interest in modern 3D microscopy to realize a purely optical position and orientation control of larger cell clusters or small embryos, i.e., of arbitrarily shaped specimen, using force-optimized grabbing positions for each optical trap. The localization of such local RI changes (see Eq. (1)) is an important topic of this study.

This localization problem resembles 3D particle tracking techniques, often using interferometric approaches and measuring phase differences between scattered and unscattered light. Important developments are summarized: The most prominent interferometric tracking technique is back focal plane (BFP)

interferometry[27], which is particularly advantageous (but not limited to) optical trapping applications and provides nanometer precision in 3D at megahertz sampling rates using quadrant photo diodes (QPDs)[28,29]. However, BFP tracking is typically limited to one particle or structure at a time (time-multiplexing)[30,31], since the signal is recorded in the Fourier plane of a detection lens (DL), where the scattered fields from all particles overlap. Usually assignments to different positions is impossible for holographic traps[16,32]. However, BFP interferometry can also be used to scan asymmetric structures, such as cell membranes[33,34] or even for shape tracking. In principle, interferometric tracking is also possible with a QPD in the focal plane[35], but requires spatial filtering for fixed focus positions for parallel trapping of several particles[36].

Most other tracking techniques based on interferometry are holographic microscopy methods using a plane reference wave. Here, a single brightfield image from a camera enables 3D position tracking of several beads (in transmission mode), which works well, if particles are well-defined and not too close to each other[37]. In reflection mode, sensitivity and the signal-to-noise ratio can be strongly increased with interferometric scattering (iScat)[38] to track several nanometer-sized particles typically in lateral directions.

In this work, we show how to sense interferometrically and in parallel local RI changes at several 3D positions within a 3D object—only by scattered laser light. Our technique, called off-focus interferometric (OFI) tracking, measures positions of particles (in general RI changes) by analyzing several interference patterns from different foci distributed over a camera. We present a local intensity gradient scheme—similar to that of QPD tracking—which we apply to beam cross-sections, which are

inherently off-focus for 3D distributions of foci. We perform advanced computer simulations to demonstrate that individual RI changes inside a focus volume can be retrieved even in the middle of clusters of scattering spheres.

## Results

Non-blind optical trapping is complicated, because on the one hand counteracting gravitational and frictional forces often out-balance the 3D optical forces. On the other hand, conventional imaging does not provide sufficient object information as feedback for the optical traps to find the optimal grabbing position, which is not automatically the edge of an object, as often assumed. This optimal position $\mathbf{b}$ is characterized by sufficiently strong optical gradient forces $\mathbf{F}_{opt}(\mathbf{b}) \approx \frac{n(\mathbf{b})}{2c} \cdot \langle \alpha(\mathbf{b}) \cdot \nabla I_i(\mathbf{b}) \rangle$. The force simplifies further to

$$\mathbf{F}_{opt}(\mathbf{b}) \approx \frac{V}{c} \cdot \langle \Delta n(\mathbf{b}) \cdot \nabla I_i(\mathbf{b}) \rangle \qquad (1)$$

if the spatial change in RI $\Delta n(\mathbf{b})$ is small, i.e., $\Delta n^2 << n^2$ such that $\alpha(\mathbf{b}) = V \cdot 2\Delta n(\mathbf{b})/n(\mathbf{b})$. $\langle \Delta n \cdot \nabla I_i \rangle$ is averaged over the center of the laser focus with volume $V$ and local intensity gradient $\nabla I_i(\mathbf{b})$[25,39] and with vacuum speed of light c.

**The setup principle.** The setup for OFI tracking requires a computer-controlled holographic trapping unit and a camera with sufficient bit-depth to record several defocused laser foci. These components were added to a commercial inverted microscope (Zeiss, Axiovert 200) and are sketched in Fig. 2a. A spatial light modulator (SLM, HoloEye, Pluto) or any other diffractive optical element generates a 3D distribution of laser foci through a water immersion objective lens (OL) (Zeiss, C-Apochromat, 40×, NA 1.2). Several objects, e.g. beads or cells in a cluster, are positioned by optical forces or other external forces within the several foci of a diffracted laser beam (Alphalas Lasers, Mono-power-1064-10W-SM, $\lambda = 1.06\,\mu m$). The laser light, scattered at multiple objects in multiple foci, is projected telecentrically by a DL (DL, W Plan-Apochromat 40×/1.0 DIC M27) and other achromatic lenses (L) onto a camera (Prosilica, GC1350H), which is conjugated to the focal plane (= reference plane $z_{ref}$, see red frame). The lateral magnification is $M = 20$, defined by the ratio of the focal lengths of L and DL. As outlined exemplarily in Fig. 2a by $N = 3$ traps at different 3D positions, $N$ laterally displaced intensity distributions $I_n(x, y, \mathbf{b}_n)$ (see orange patches) are recorded by the camera ($n \leq N$). Hence, the camera records N focal spots at axial positions $z_{dn} \cdot M^2$, which are typically all off-focus. The rectangular inset on top defines the particle's 3D trapping position $\mathbf{b}_n = (b_x, b_y, b_z)_{n \leq N}$ relative to each laser focus, with center position $\mathbf{t}_n$ relative to the center of the reference FP plane (Fig. 2b). The scattered laser light (indicated by orange waves) and the intensity cross-sections $I_n(x, y, \mathbf{b}_n)$ on the camera change with each position of the particle or the structure. The brightfield image shows a 170 μm large cancer cell cluster (with ca. 2000 cells, each 10–15 μm in diameter) held in the focal plane by $N = 8$ optical traps (indicated by green cross-markers).

**Theoretical concept.** In this section, we address the problem of how to estimate the trapping strength of multiple structures in multiple foci with a single camera shot. In particular, we describe how local RI changes $\Delta n(\mathbf{b}_n)$ and thereby optical forces (see Eq. (1)) can be read out by scattered laser light. The interference intensity distribution on a plane at $z_d$ conjugated to the camera plane $I_d(\mathbf{b}_1, \mathbf{b}_2, ..) \approx I_{d1}(\mathbf{b}_1) + I_{d2}(\mathbf{b}_2) + ...$ is the sum of several, non-overlapping off-focus laser beam cross-sections, all with different 3D center positions $(\mathbf{t}_1, \mathbf{t}_2, ..)$ and relative positions $(\mathbf{b}_1, \mathbf{b}_2, ..)$ of $N$ particles as sketched in Fig. 2a, b. Hence, the intensity of each beam $I_{dn}(\mathbf{r}_d, \mathbf{b}_n + \mathbf{t}_n)$ depends on the 3D

positions of the camera $\mathbf{r}_d = (x, y, z_d)$, the relative particle position $\mathbf{b}_n$ and the trap center position $\mathbf{t}_n$.

**Interference intensities.** In our simplified model, scattering between the particles is neglected, i.e., $|\sum_{n=1}^{N}(E_{in} + E_{sn}(\mathbf{b}_n))|^2 \approx \sum_{n=1}^{N}|E_{in} + E_{sn}(\mathbf{b}_n)|^2$, which can be achieved for $N < 10$ and an adequate camera position at $z_d$. Figure 3a illustrates the fields $E_i(x, 0, z) + E_s(x, 0, z, \mathbf{b})$ of a single beam focused with NA. It further indicates a distance $z_{dn}$ behind the geometrical focus (i.e., at a distance $M^2 \cdot z_{dn}$ at the conjugate camera plane). The incident electric field $E_i$ (including wavefronts) is shown in blue, the scattered electric field $E_s(\mathbf{b})$ from a point scatterer at position $\mathbf{b}$ (with spherical wavefronts) is sketched in green. The resulting off-focus interference (OFI) intensity distribution $|E_i(x, y) + E_s(x, y, \mathbf{b})|^2 = I_i + I_s(\mathbf{b}) + I_{is}(\mathbf{b})$ results in three intensity terms for unscattered, scattered and interference light, $I_{is}(\mathbf{b}) = 2\text{Re}\{E_i^* E_s(\mathbf{b})\}$. Hence, the total intensity of N deformed beams at the camera in the (conjugate) plane $z_d$ reads

$$\begin{aligned} I_d(\mathbf{b}_1, \mathbf{b}_2, ..) &\approx \sum_{n=1}^{N} I_{dn}(\mathbf{b}_n) \\ &= \sum_{n=1}^{N}\left(I_{in} + I_{sn}(\mathbf{b}_n) + I_{cn}(\mathbf{b}_n)\sin(\Delta\phi(\mathbf{b}_n))\right) \end{aligned} \qquad (2)$$

with $I_{cn}(\mathbf{b}) = 2\sqrt{I_{in} \cdot I_{sn}(\mathbf{b})}$ and $\sin(\Delta\phi(\mathbf{b}) = \cos(\Delta\phi(\mathbf{b}) + \frac{\pi}{2})$ for particles with a RI $\Delta n(\mathbf{b})$ higher than that of the environment ($n_m$).

**Local phase shifts.** The position-dependent difference between the k-vectors of the incident and the scattered wave, $\Delta\mathbf{k}(\mathbf{b}) = \mathbf{k}_i - \mathbf{k}_s(\mathbf{b})$, defines the local phase difference $\Delta\phi(\mathbf{b}, \mathbf{r}_d) \approx \Delta\mathbf{k}(\mathbf{b}) \cdot \mathbf{r}_d + \theta_s$ in the plane conjugated to the camera. For particles within the Born-approximation, the phase-shift $\theta_s$ vanishes ($\theta_s = 0$), whereas $\theta_s > 0$ for larger particles.

The interference intensity $\approx I_c(b_x, b_y, b_z) \cdot \sin(\Delta\phi(b_x) + \Delta\phi(b_y) + \Delta\phi(b_z))$ between the incident focused and the scattered wave (both diverging with the NA of the lens) changes also for small axial displacements $b_z$ with the Gouy phase difference $\Delta\phi(b_z) = \text{atan}(k_0 b_z NA^2/n_m) \approx (k_0 NA^2/n_m) \cdot b_z$ The Gouy phase shift is inherent in every non-collimated beam and is also essential for 3D BFP tracking[29].

The three intensity profiles $I_i(x)$, $I_d(x)$ and $I_{is}(x) \approx I_d(x) - I_i(x)$ at $z_d = 20\,\mu m$ along x are sketched in Fig. 3b, an analytical calculation of the 2D intensities $I_d(x, y, z_d, \mathbf{b})$ and $I_{is}(x, y, z_d, \mathbf{b})$ based on an incident Gaussian beam and a laterally displaced Rayleigh scatterer at $\mathbf{b} = (0.25, 0, 0.1) \cdot \mu m$ is shown in Fig. 3c, d. The scattering strength was approximated by the Clausius-Mossotti polarizability for a 1 μm glass bead with $n_s = 1.47$. The vacuum wavelength was $\lambda = 1.06\,\mu m$, the RI of water $n_m = 1.33$ and the numerical aperture of the focusing lens NA = 1.2.

**Integrated position signal.** A one-dimensional position signal, e.g., $S_{xn}(\mathbf{b})$ in x-direction, for relative particle displacements $\mathbf{b}$ can be analyzed by integrating the intensity $I_d(x, y, \mathbf{b}) \approx I_i(x, y) + I_{is}(x, y, \mathbf{b})$ of each beam over an appropriately chosen circular area $\Omega_n$ (Fig. 3b–d) to find the center-of-intensity signal according to

$$S_{xn}(\mathbf{b}, z_{dn}) = \frac{1}{\bar{S}_n(\mathbf{b}, z_{dn})} \cdot \frac{1}{z_{dn}} \int_{\Omega_n} x \cdot I_{dn}(x, y, z_{dn}, \mathbf{b}) dxdy \qquad (3)$$

where $I_{dn}(x, y, z_{dn}, \mathbf{b}) \approx I_{cn}(x, y, z_{dn}, \mathbf{b}) \cdot \sin(\Delta\phi(x, y, z_{dn}, \mathbf{b}))$. $\bar{S}_n(\mathbf{b}, z_{dn}) = \int_{\Omega_n} I_{dn}(x, y, z_{dn}, \mathbf{b}) dxdy$ is the average power in the n-th beam and with zero mean position of the incident light $\int x \cdot I_{in}(x, y) dxdy \approx 0$. The y-position signal $S_{yn}(\mathbf{b}, z_{dn})$ is obtained correspondingly.

The axial z-position signal $S_{zn}(\mathbf{b}, z_{dn})$ in the n-th beam can be obtained by subtracting and normalizing by the intensity $\bar{S}_{0n}(z_{dn})$

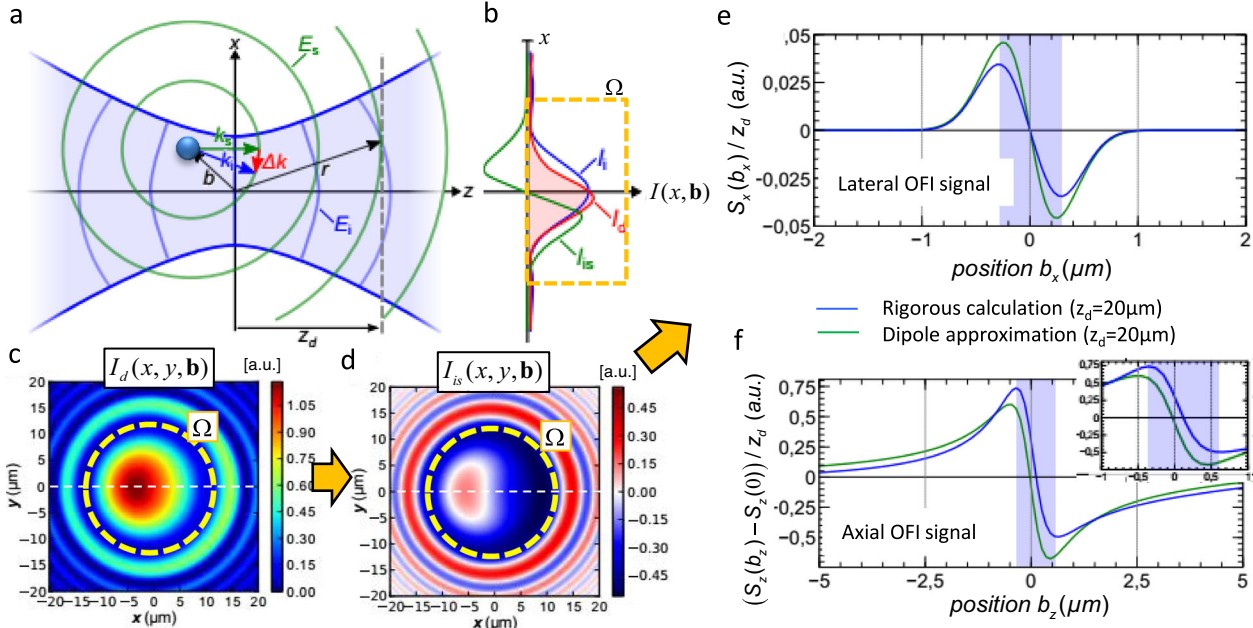

**Fig. 3 Calculating interference intensities at the camera plane. a** Scattering scheme: A spherical wave with field $E_s$ (green) is emitted by a point scatterer at position b relative to the center of a Gaussian laser focus with field $E_i$ (blue). The phase difference is proportional to the difference of the momentum vectors $\Delta k = k_i - k_s$. **b** One-dimensional intensities $I_i(x)$ without scatterer and $I_d(x) \approx I_i(x) + I_{is}(x)$ with scatterer at plane $z_d$ behind the focus, with $I_{is}(x)$ being the interference intensity. The dashed line indicates the circular integration area $\Omega$. **c, d** Normalized two-dimensional intensities $I_d(x,y,b)$ and $I_{is}(x,y,b)$ at plane $z_d$ obtained from a scattering simulation with a glass bead at b = (0.25,0,0.1)·μm. **e, f** The lateral (**e**) and axial (**f**) position signals, $S_x(b)$ and $S_z(b)$, are obtained for positions $b_x$, $b_z$ along two axes through the focus after calculating the center-of-intensity from $I_{is}(b)$. The results from a simplified analytical dipole model (green) and a rigorous numerical simulation (blue) are compared.

without a particle, such that

$$S_{zn}(\mathbf{b}, z_{dn}) = (\bar{S}_n(\mathbf{b}, z_{dn}) - \bar{S}_{0n}(z_{dn}))/\bar{S}_{0n}(z_{dn}) \qquad (4)$$

Based on the equations above, the position signals in lateral and axial direction, $S_x(b_x, z_d)$ and $S_z(b_z, z_d)$, for particle displacements along the focus center were calculated for an incident focused Gaussian beam $I_i(\mathbf{r})$ and a Rayleigh particle (see parameters above). As displayed in Fig. 3e, f, the blue curve represents a rigorous calculation, whereas the green curve represents the paraxial approximation of the interference term $I_{is}(\mathbf{r}, \mathbf{b})$. The shape and amplitude of both position signals are in a good qualitative agreement and reveal a linear course based on $I_{is}(\mathbf{r}, \mathbf{b}) = I_c(\mathbf{b}) \sin(\Delta\phi(\mathbf{b}))$, demonstrating the validity of our simplified analytical model. The unique tracking range is shaded in blue, the central region of the axial position signals is magnified in the inset. The linear range is well visible, i.e., where the tracking signal $S_j(b_j, z_d) \approx g_j \cdot b_j$ is proportional to the displacement in direction $j = x, y, z$ by a calibration factor $g_j$.

The basic requirement for correct instrument alignment and precise experiments is that the OL and the DL have a confocal alignment (common focal plane), which is the object plane (reference plane at $z_{ref} = 0$) and that the pupil planes of both lenses ensure collimated illumination and detection, respectively.

**Object rotation with axis in image plane.** In an initial experiment, we examined whether the optical forces generated by holographic optical tweezers are strong enough to lift or rotate large objects. Using dynamic computer holograms generated by a SLM, we moved three, arbitrarily distributed optical point traps across a spherical surface, to rotate a biological specimen (with rotation axis in image plane), without knowing the optimal grabbing positions. As illustrated in Fig. 1b, we could exert optical torques to a 70 μm large RH30 cell cluster by the three point traps, leading to a full $2\pi$ rotation within 140 s. A cell cluster, 10

times larger in volume and weight, could be lifted from the coverslip through eight optical traps (Fig. 2a). However, it could not be rotated, since local gradient forces were yet not efficient enough to overcome friction.

**OFI position tracking of beads.** The intensity distributions on the camera and the resulting position signals predicted by the theoretical model were verified in a first type of experiment using isolated particles. $N = 6$ optical traps were distributed equally across a spherical surface with 70 μm in diameter (Fig. 4a). This diameter corresponds approximately to the commonly used size of cancer cell clusters. The six optical foci were positioned equidistantly in 3D space, such that their off-focus detection distances $z_{dn}$ cover an axial range of nearly 70 μm, i.e., with distances $z_{dn} = -34, -20, -7, +7, +20, +34$ μm to the reference plane at $z_{ref} = 0$. The resulting beam cross-sections $I_{dn}(x, y, z_d, \mathbf{b}_n)$ (Fig. 2a), each with a different diameter $2 \cdot R_{sn}$, then still have a sufficiently large distance to each other to avoid significant beam overlap. For a water immersion DL with NA = 1.33· sin$\alpha$ = 1.0, the detection angle is $\alpha < 47°$, such that the $n$-th spot radius $R_{sn} = M^2 \cdot \tan\alpha \cdot z_{dn} \approx M^2 \cdot z_{dn}$ is approximately equal to the magnified distance $M^2 \cdot z_{dn}$ of the $n$-th beam. A distribution of $N = 6$ beam cross-sections is shown in Fig. 4b, which is the logarithmic intensity of the camera image, overlaid with 6 circular analysis regions $\Omega_n$.

Parallel trapping of six glass beads with RI $n_s = 1.47$ and 1 μm in diameter was easily possible, which has become a standard task over the last decade. While stable optical trapping is not a big challenge, it is more difficult to achieve precise, linear and orthogonal particle tracking in 3D. Therefore, we measured the detector responses for orthogonal particle displacements through each of the six laser foci in a meander-like scan as outlined in Fig. 4a. This was achieved by attaching the glass beads to the coverslip and moving the 3D piezo-stage (MCL, Nanoview300)

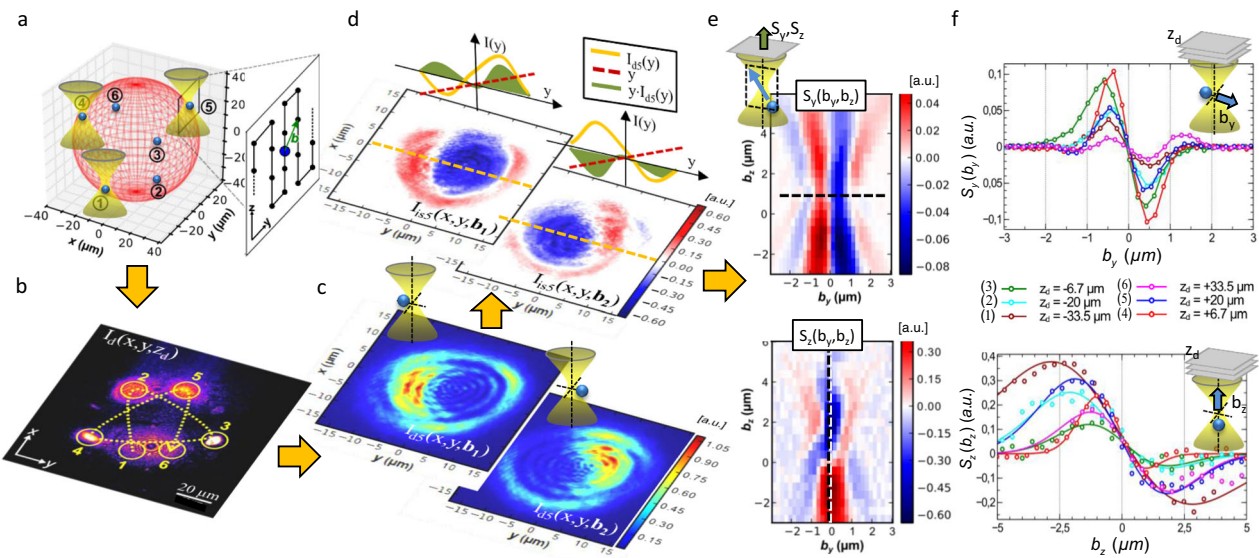

**Fig. 4 Generation of 3D position signals of 6 particles inside 6 optical traps. a** Particles and optical traps are distributed across a spherical surface with 80 μm in diameter (in red). Particles at position b (green arrow) move within each focal region, e.g., externally driven in the yz-plane in meander scan mode. **b** Snapshot at a camera plane conjugated to $z_d = 20$ μm behind the reference focal plane showing (logarithmic) intensity distributions of six beams deformed by 6 trapped glass beads. **c** Two exemplary intensity distributions $I_d(x,y,b)$ at $z_d = 20$ μm for positions b = (±0.25, 0, 0.1)·μm of a 1 μm glass bead displaced in trap $n = 5$. **d** Corresponding 2D interference distributions $I_{is}(x,y,b)$ and schematic intensity profiles $I(y)$ (orange) to obtain the center-of-intensity $\iint y \cdot I_{dn}(x,y)dxdy$. **e** Lateral position signal map $S_y(b_y,b_z)$ (top) and axial position signal map $S_z(b_y,b_z)$ (bottom) for bead positions $(b_y,b_z)$ within the focal region. **f** Position signal linescans in y-lateral (top) and z-axial (bottom) direction through the focus for different camera positions $z_d = -33$ μm, …,33 μm. Position signals (markers) with fit functions (lines) are approximately linear with the particle displacements over the extent of the focus.

with coverslip and bead in controlled steps of 200 nm laterally and 400 nm axially.

To illustrate the optical principle and to allow comparisons to the theoretical model, Fig. 4c shows a magnification of the intensity area $I_{d5}(x,y,z_{d,5},b)$ of the $n = 5$th trap with axial displacement $z_{d5} = 20$ μm and lateral bead displacements b = (±0.25, 0, 0.1)·μm. By subtracting the intensity distributions $I_{i5}(x,y,z_{d,5})$ without particle, the pure intensity changes for particle displacements b = (±0.25, 0, 0.1)·μm become visible in the two plots of Fig. 4d. This is further illustrated in the Supplementary Movie 4. The dashed lines mark the schematic intensity profiles $I_{d5}(y,b)$ (orange) and the unity line (red), which (after multiplication) results in the greenish area $\int y \cdot I_{d5}(y,b)\, dy$, which is proportional to the OFI-signal. Applying this procedure, described in Eqs. (3) and (4), results in the two-dimensional signal plots $S_y(b_y,b_z)$ and $S_z(b_y,b_z)$ of Fig. 4e, where positive signals are shown in red and negative signal values in blue. In a region of 6 μm × 9 μm particles were scanned in the $(b_y,b_z)$-plane through the stationary focus (at $b_x = 0$). A horizontal line scan $S_y(b_y,z_d)$ through the top plot and a vertical line scan $S_z(b_z,z_d)$ through the bottom plot for $z_d = +20$ μm were performed and are displayed in blue colors in Fig. 4f. The same signal scans were performed for the other five particles in the five other laser foci corresponding to different off-focus positions $z_d$. The corresponding line scans are shown in the two graphs for lateral and axial particle displacements in Fig. 4f. All experimental data (circular markers) can be well fitted by functions of type $A \cdot (C - b_j) \cdot \exp(-b_j^2/B^2) + D$, and only differ in amplitude and width, i.e., in sensitivity $\frac{\partial}{\partial b_j}S_j(b_j,z_d)$ and linear range around $b_j = 0$ ($j = x,y,z$).

**OFI localization and mapping inside a cluster of beads.** When a plane wave propagates through a cluster of beads, the phase

perturbation $\delta\varphi(b, \Delta z)$ of the wave increases approximately linearly with propagation distance $\Delta z$[40]. However, this is different for a focused wave, when the sphere size and phase perturbation increase relative to the decreasing beam diameter. We performed computer simulations using the beam propagation method (BPM)[40] to investigate whether it is possible to measure the characteristic phase change $\Delta\phi(b)$ in the focus (see Eq. (2)) induced by a single 1.1 μm sized glass sphere, which is surrounded by hundreds of other spheres, generating a phase shift $\delta\varphi(b, \Delta z)$. To be able to detect the single sphere of interest, it should be $|\delta\varphi(b, \Delta z)| << |\Delta\phi(b)|$, such that the resulting intensity changes $\frac{\partial}{\partial b}I_{is}$ on the camera to do multiple scattering are small, i.e.

$$\frac{\partial}{\partial b}(I_c(\mathbf{b}) \cdot \sin(\Delta\phi(\mathbf{b}) + \delta\varphi(\mathbf{b}, \Delta z))) \approx \frac{\partial}{\partial b}(I_c(\mathbf{b}) \cdot \sin(\Delta\phi(\mathbf{b}))).$$

(5)

This computer experiment is illustrated in Fig. 5a where each NIR-laser beam focused at a scan position $\mathbf{b} = (b_x, b_y, b_z)$ generates diffraction patterns $I_d(x,y,z_d,\mathbf{b})$ on a CCD camera in a defocused position with, e.g., $z_d = -4$ μm. Two such off-focus camera signals from beam positions $(b_{x0}, b_{y0})$ and $(b_{x1}, b_{y1})$ are shown in green and orange frames. We performed more than 15,000 BPM simulations for different focus positions within a 50 μm cluster and for different detector positions. Figure 5b and Supplementary Movie 6 show that each single bead, i.e., each local RI change (within the four regions of interest) can be reproduced well by the OFI beam scans displayed in pseudo-colors on the right. Both wavefronts, before and behind the bead of interest are heavily phase distorted, but result in a good, edge-enhanced image of the bead from the incoherent superposition of OFI signals $S_\perp(b_x, b_y)$ and $S_\perp(b_x, b_z)$, where $S_\perp = \sqrt{S_x^2 + S_y^2}$.

**OFI localization and mapping of cellular structures.** A spherical bead represents a structure with well-defined shape and RI

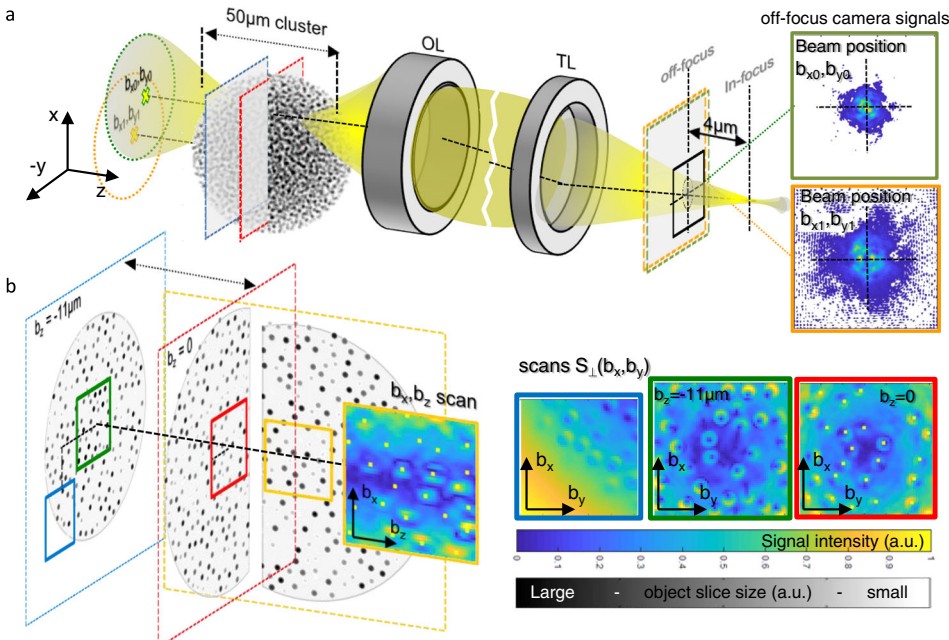

**Fig. 5 Simulated OFI scanning through inhomogeneous media. a** A focused beam at position $(b_{x0}, b_{y0})$ (or $(b_{x1}, b_{y1})$ respectively) propagates (BPM) through a 50 µm large cluster of 4900 1.1 µm spheres. The coherently scattered light is imaged by the objective lens (OL) and tube lens (TL) onto a camera, which is placed by $z_d = -4$ µm before the focus (frames with dashed lines). Two exemplary diffraction images $I_{d0}(x, y, z_d, \mathbf{b}_0)$ and $I_{d1}(x, y, z_d, \mathbf{b}_1)$ are shown on the right (green and orange frames). Two exemplary scan planes for axial scan positions $b_{z1} = 0$ and $b_{z2} = -11$µm are indicated by a red and blue frame within the cluster. **b** OFI beam scans within four different planes inside the sphere cluster. The orange frame is from an OFI scan in the xz plane, the other three frames are from OFI scans in different xy planes. Each scan consists of 60 × 60 beam positions (from each 3600 BPM simulations) revealing clear images of individual spheres at their correct positions (yellow dots). The shaded gray background on the left indicates the cross-section of the sphere cluster.

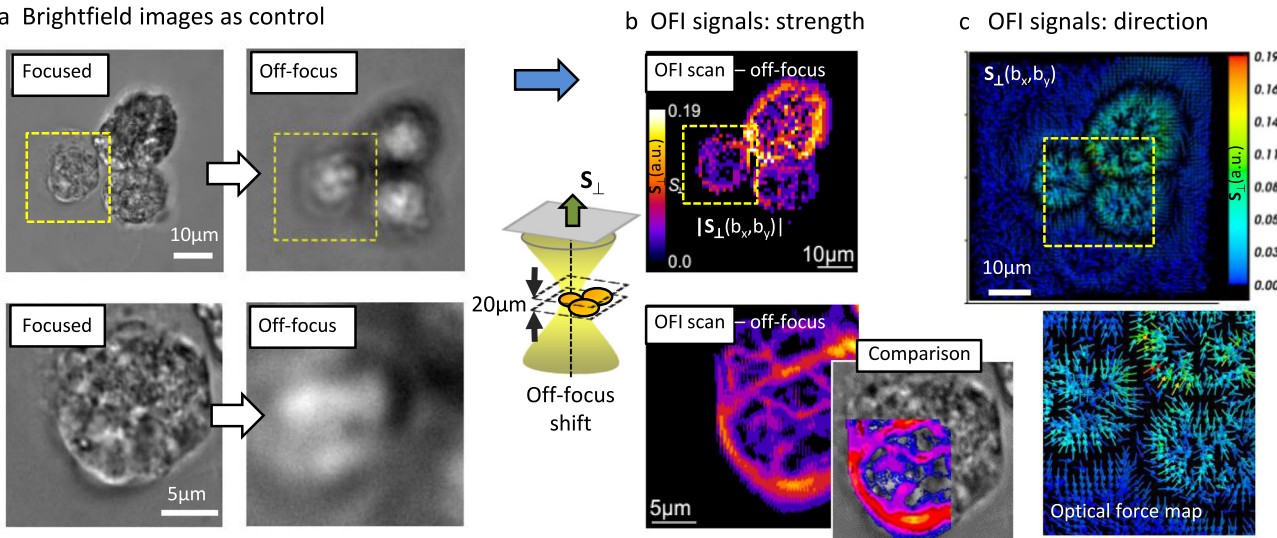

**Fig. 6 Conventional imaging and OFI scanning. a** Conventional brightfield imaging of three RH30 cells in the focal plane (focused, for comparison) and $z_d = 20$ µm above the focal plane (off-focus). Regions of interest (yellow area) are magnified in the bottom line. **b** Off-focus interferometric (OFI) scan | $S_\perp(b_x, b_y)$| of cells at $z_d = 20$ µm provides detailed phase image of cells representing refractive index changes. Scan steps: 1 µm (top) and 0.5 µm (bottom). For comparison, the OFI scan has been partly overlaid with the focused image. **c** Vector plot $S_\perp(b_x, b_y)$ of OFI scans reveals the cells refractive indices gradients $\nabla n(x, y)$ or optical forces $F_{opt}(x, y)$ with directional change at the cell membranes (black areas). These experiments were repeated with similar results ($N > 5$).

change $\delta n(\mathbf{b})$, which can diffuse in a homogeneous or more complex environment. A cellular structure is usually not well-defined, neither in shape nor in $\delta n(\mathbf{b})$, and movements within cellular environments are relatively slow. However, it is the change in RI $\delta n(\mathbf{b}) \propto \mathbf{F}_{opt}(\mathbf{b})$ at position $\mathbf{b}$ that generates contrast,

e.g., in brightfield imaging with intensity $I_d(x, y, z_d, \delta n(\mathbf{b})) = |E_i(\mathbf{r}) + E_s(\mathbf{r}, \delta n(\mathbf{b}))|^2$ and that needs to be determined to find the optimal optical trapping forces $\mathbf{F}_{opt}(\mathbf{b})$. Figure 6a illustrates that brightfield images (using a non-coherent light source) provide a

reasonable distribution of refractive indices in the focal plane[39], but not for off-focus planes (defocussed by $z_d = 20\,\mu m$).

Here, we demonstrate in a second type of experiments that OFI scanning allows to acquire also high-resolution maps of refractive indices although the trap is clearly off-focus ($z_d = 20\,\mu m$). First, we scanned a small cluster of unlabeled RH30 cells attached to a coverslip through one of the laser foci. The cross-section of the distorted beam on the camera was analyzed according to the steps explained in the theory section to obtain the signal maps $S_x(b_x, b_y) = S_x(\delta n(b_x, b_y))$ and $S_y(b_x, b_y) = S_y(\delta n(b_x, b_y))$. These OFI signals describe the change in RI of the cells in x and y direction, respectively. However, a full vectorial information about the local RI changes can be obtained by applying the following signal operation

$$S_\perp(\delta n(\mathbf{b})) = \sqrt{S_x^2(\mathbf{b}) + S_y^2(\mathbf{b})}, \, \mathbf{S}_\perp(\delta n(\mathbf{b})) = \begin{pmatrix} S_x(\mathbf{b}) \\ S_y(\mathbf{b}) \end{pmatrix}, \quad (6)$$

Here, $S_\perp(\delta n(\mathbf{b}))$ describes the strength of the RI change $\delta n(\mathbf{b})$ at position $\mathbf{b}$ and the vector $\mathbf{S}_\perp(\delta n(\mathbf{b}))$ both strength and direction of $\delta n(\mathbf{b})$ or the gradient $\nabla n(\mathbf{b})$, respectively. The strength of the OFI signals of the three cells is mapped in Fig. 6b in pseudo colors, where the top row shows an image scanned in 1 µm steps, while the bottom row shows a magnification of the yellow ROI scanned in 0.5 µm steps. The near-infrared OFI scan is compared with the off-focus brightfield image image by the false-color overlay. Correspondingly, Fig. 6c shows two vector plots $\mathbf{S}_\perp(\delta n(\mathbf{b}))$ of the same scenario, where arrows indicate the directional change and pseudo colors indicate the strength of the OFI signals. In both representations of OFI scans, the positions of the cell membranes and the contact areas can be well recognized, which is impossible with off-focus brightfield imaging. Since the OFI signal vector is approximately proportional to the gradient of the RI, it is also proportional to the optical gradient force assuming an unchanged focus volume as introduced in Eq. (1). Hence, the following relation represents a directional map of OFI signals, RI gradients, and optical forces for each position ($b_x$, $b_y$):[41]

$$\mathbf{S}_\perp(\mathbf{b}) \sim \iint \begin{pmatrix} x \\ y \end{pmatrix} \cdot I_d(x, y, \delta n(\mathbf{b})) dx dy \sim \nabla n(\mathbf{b}) \sim \mathbf{F}_{opt}(\mathbf{b})$$

$$(7)$$

Figure 6 shows that OFI scanning does not require refocusing over a large axial range ($z_d = 20\,\mu m$). In consequence, it must be possible to perform OFI scans of cell clusters also in differently oriented planes or along arbitrary trajectories opening the option to scan along specific volume areas of interest, such as the edges of cells at the periphery of the cell cluster.

In a third type of experiment, a 70 µm large cell cluster remained stationary and three laser foci—now displaced by continuously varying holograms of the SLM—scanned three cells at the edge of the cluster as illustrated in Fig. 7a. The brightfield image shows a central plane of the cell cluster, such that only some cells appear focused, whereas the majority of cells are defocused and blurred. Three OFI scans sampled in a 0.5 µm raster were performed by three lasers in a 27 µm × 27 µm region tilted by 45°, as indicated by the projected red ROIs c, d, e together with the corresponding pseudo-color OFI maps. These three maps $S_\perp(\mathbf{b}) = \sqrt{S_x(\mathbf{b}) + S_y(\mathbf{b})}$ are magnified in the second row of Fig. 7. It can be seen that the cell of ROI c is strongly out of focus and any details of cellular structures are invisible. However, with the object cross-section performed by the OFI-scan, regions with large RI changes at the cell periphery can be detected (although less precise), such that optical tweezers can apply forces at this position.

## Discussion

The motivation of this study was to evaluate a technical approach based on optical forces that allow to flexibly hold and rotate biological specimen of a few 100 µm in size for taking 3D images (e.g., with light-sheet microscopy)—without using coverslips or gel cylinders for object mounting. It has been shown in several studies that single cells can be well rotated in three directions using optical tweezers at laser powers of a few milliwatts. However, larger specimen, consisting of hundreds or thousands of cells are difficult to manipulate because of increased friction and gravitational forces—especially for rotations out of the image plane and for blind optical forces (Fig. 2c).

In a first necessary proof of principle step, we have shown for the first time that optical rotation of a 70 µm large RH30 cancer cell cluster around the x-axis, i.e., out of the image plane is possible within 140 s using 3 optical traps (Fig. 1b and Supplementary Movie 3). Because of the 5–10 fold weaker axial intensity gradients, this is more difficult than rotation within the image plane, i.e., around the optical axis. Stable optical lifting and holding was possible even for a 10-fold larger cell cluster (170 µm in diameter) with 8 optical traps of each 56 mW laser power (Fig. 2a and Supplementary Movie 5). Although the high laser powers will eventually kill the few cells being constantly in the same laser focus during rotation, we assume the investigation of the other cells and the whole specimen is hardly affected by this. But, in both experiments, the optical traps have been distributed nearly equally across a spherical surface by a simple blind control algorithm, i.e., without probing the optimal grabbing position for each trap. It can be seen that the optical traps slip off repetitively during rotation. Our approach—"OFI localization"—aims to solve such problems and to lay a foundation for non-blind optical trapping.

What is OFI localization of scatterers? And how does it compare to known techniques for imaging or tracking?

The OFI principle is different to BFP tracking[28,29], since OFI records phase changes from particles or structures in the focal plane region. The structure with a RI different from the environment modulates the phase depending on the wavelength, the degree of coherence, and the degree of focusing. Therefore, regular widefield phase-contrast microscopy methods provide phase information that has only limited relevance to the trapping focus. OFI localization is based on a focused laser beam, which scatters coherently at a structure at position $\mathbf{b}$, leading to a distorted beam intensity $I_n(x, y, z_d, \mathbf{b})$, which is analyzed with a camera at an off-focus position $z_b$. The camera records images of typically $N = 5$–10 deformed laser beams at the same time (after passing a 4f imaging system) in distances $z_d = \pm 5\,\mu m \ldots \pm 50\,\mu m$ away from the laser focus, from which a center-of-intensity signal $\mathbf{S} = (S_x, S_y, S_z)$ is processed and stored in x, y and z direction. By processing the center-of-intensity $\int \binom{x}{y} I_n(x, y, \mathbf{b}) dx dy$ obtained by the camera, a non-linearity is introduced to the process of signal generation, which resembles a derivative operation $\frac{\partial}{\partial b}$ along x- or y-direction (Fig. 4d). This non-linearity can also be introduced by a detector, such as a position sensitive device (PSD) or a QPD. Remarkably, the OFI- principle to detect RI changes works also in a highly scattering environment, as we have shown by the results of advanced computer simulations (BMP method) shown in Fig. 5. Using a 100 Hz camera and standard CPU or GPU operations, $N$ x100 scattering patterns from $N$ beams could be analyzed to find the optimal beam deflection and grabbing position, which will be one of the next steps to do.

There are two typical working modes: (i) OFI is used in tracking mode, when small structures move within the stationary laser focus volume. (ii) OFI is used in imaging mode, when the laser focus is scanned across a stationary structure.

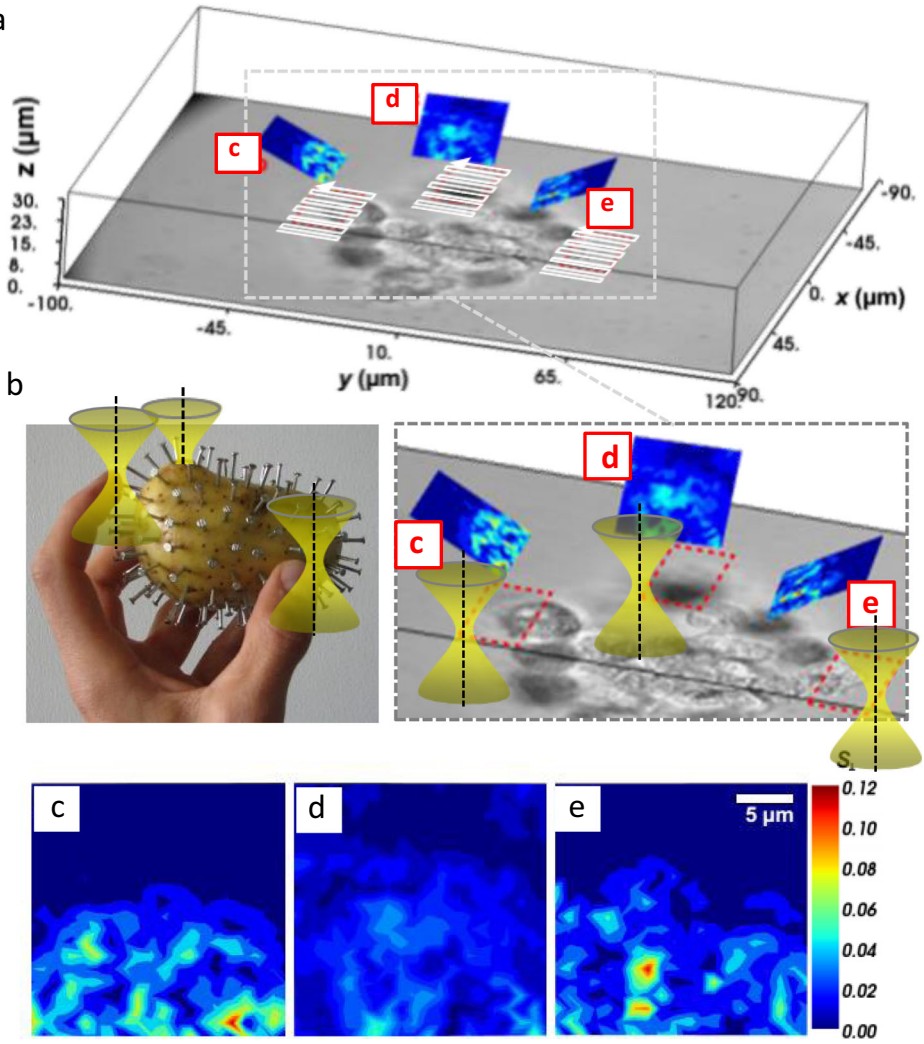

**Fig. 7 Oblique OFI scanning of cells within a cell cluster. a** Brightfield image (in grayscale) of one plane within the cell cluster combined with three, differently oriented, oblique planes c, d, and e at the edges of individual cells (in pseudo colors). **b** The fingertips have to find the nailheads, i.e., the most efficient grabbing position to to apply a controlled torque to the potato with nails. **c–e** Strengths $|S_\perp(b)|$ of the lateral OFI signal map of a $27 \times 27$ µm region tilted by 45° were obtained by scanning three laser foci across a stationary cell cluster. The edges of the cell and thereby the potentially best positions for optical traps to grab are visible, even for the completely blurred cell in ROI d.

In the OFI tracking mode for a static focus, the time trajectories $S(t)$ of the same small structure having a constant RI (e.g., a bead) are recorded. OFI tracking combines the advantages of holographic video tracking[37] and BFP interferometric tracking[28,29]. The first technique uses a planar reference wave and a camera and hence can track several particles in parallel. The second uses a focused laser beam and a QPD, but does not require the knowledge of any optical properties of the particle, especially when thermal noise calibration is applied. As shown in Fig. 3f, the linear detection range ($\sim\lambda$ laterally, 2–4 $\lambda$ axially) is of similar length as in BFP tracking, provided that the defocused distance $z_d$ is not too small (as illustrated by the red curve for axial signals). Although larger off-focus distances provide longer linear detection ranges (brown curve for axial signals), this goes along with larger cross-sections on the camera, leading to an overlap between adjacent beams, which can complicate postprocessing and analysis. However, as shown by the adjacent positions of optical traps $n = 1$ and $n = 6$ in Fig. 3b, the beam overlap hardly influences the slope of position signals in Fig. 3f. It is further noticeable that the signal sensitivities (slopes) of the lateral and axial detection signals, $S_{xy}$ and $S_z$, change inversely with the beam diameter, which is known from BFP tracking[42]. Here, analyzing larger (smaller)

beam cross-sections does improve linearity and sensitivity for lateral (axial) tracking. This would be also one of the future steps to go with OFI tracking.

Hence, both the position of the camera plane and the distribution of laser foci in 3D space should be chosen such that defocused spots have a radius $R_s > M^2 \cdot 10$ µm (Fig. 3c, f) and do not overlap (at magnification $M$). This enables a sufficiently precise and flexible tracking of several particles in parallel, when computer-holographic focus control is favored, relative to time-multiplexed approaches[23]. The holographic trap control brings us to the second OFI mode, which can be used in connection with the multiple tweezers based rotation of cell clusters and other biological objects, which are often larger than 100 µm.

In the OFI imaging mode for static objects, the position signals $S(b, z) \approx S(b, z + z_d)$ of structures varying in RI (e.g., cells) are recorded and mapped to an image, where signals are nearly independent of $z_d$. A confocal scanning microscope, for comparison, typically collects incoherent fluorescence light at each focus position using sensitive point detectors, such that the fluorescence intensities $I_{fluo}(b)$ are mapped to an image.

OFI uses coherent light from a single laser, which propagates through inhomogeneous material, suffering from multiple phase

perturbations, and measures a differential signal. Whereas the phase front of a beam with a large cross-section is only slightly distorted by the scatterers, the same type of scatterer in the beam's focal region (minimum cross-section) generates a large phase perturbation and a stronger redirection of momentum vectors (k-vectors as the local gradient of the phase front). In consequence, the spatial change in beam intensity is more pronounced, which can be well detected on an off-focus camera plane as illustrated by the computer experiments in Fig. 5. The OFI technique is rather robust against multiple scattering before and behind the object of interest, although based on coherent laser light. This is the reason why RI changes $\nabla n(\mathbf{b})$ inside inhomogeneous objects, such as cell clusters, can be mapped well by OFI point-scanning as demonstrated in Figs. 6 and 7.

In Fig. 6b, c we could show that the OFI signal triplet $\mathbf{S}(\mathbf{b})$ can be plotted as a vector, displaying both the strength and the direction of the RI changes $\nabla n(\mathbf{b})$ or the optical forces, respectively. Again, the gradient can be retrieved by spatial differentiation from processing the center-of-intensity. Since the mapped center-of-intensity for each scan point does not change much with the off-focus distance $z_d$, also oblique scans can be performed without requiring any refocusing as illustrated in Fig. 7.

The OFI approach is related to direct optical force measurements based on a shift of the center-of-intensity position[41,43,44], which is however typically measured in the BFP. Here, the reflection of the mean momentum vector permits direct measurement of the momentum transfer, and hence the force, applied to the trapped object—assuming that all the scattered light can be collected. Although measuring absolute optical forces would be another interesting future investigation, the main purpose of this study was to identify a strategy to detect relative changes in optical forces as described by relation (7), where $\mathbf{S}_\perp(\delta n(\mathbf{b})) \sim \nabla n(\mathbf{b}) \sim \mathbf{F}_{opt}(\mathbf{b})$. The maximum amplitudes in the force gradient maps of Fig. 6b or equivalently the dark (zero-force) regions in the directional plots of Fig. 6c indicate the expected most efficient positions for the optical traps to grab the cells (optical potential minima).

Further scanning experiments with different laser illumination wavelengths need to be performed to investigate the influence on scattering strength, image contrast and image resolution. Alternative approaches to apply OFI also for high-quality 3D coherent imaging might be to vary, e.g., the illumination angle and the degree of coherence[45]. OFI imaging might have potential applications in fast object screening of especially biological specimen, consisting of hundreds to thousands of cells. Here, OFI could provide a distribution of refractive indices (i.e., of cellular material) relative to the fluorescence distribution, which could be imaged in parallel.

Motivated by the vision to stably hold, move, and 3D orient large biological specimen such as cell clusters and small embryos in the future purely by optical forces, we have gone an important step in this direction with the present study. We have shown that by OFI label-free imaging of RI changes is amazingly robust against multiple scattering, such that this technique is not only useful for the localization of grabbing positions for optical traps, but also for 3D imaging. As the basic requirement for this long-term tweezing challenge, we demonstrated on the one hand the optical force driven out-of-focal plane rotation of a 70 µm large cell cluster, and on the other hand single off-focus camera shots to recover the 3D gradient force distribution within cell clusters. Our future goal is to let multiple feedback-controlled laser foci find their optimal grabbing positions at positions of high RI gradients on their own—similar to our fingers trying to rotate a very fragile or spiky object, with each finger continuously sensing

the pressure applied to the object (Fig. 7b). We want to realize our vision through intelligent gradient search algorithms, multiple and cross-correlated feedback OFI data analysis for local and global stability of optical potentials. In connection with deep-learning optimization approaches and—as discussed above—a further improved OFI tracking and imaging, it should be possible that one optical trap exerts torque and force without destabilizing the other traps, while rotating a specimen to an arbitrary orientation.

## Methods

**Sample preparation**. RH30 cancer cells, RH30 human rhabdomyosarcoma (DSMZ Braunschweig – ACC 489) were cultivated in RPMI 1640 medium with 10% FCS (fetale bovine serume, Biochrom). Cells growing adherently are typsinated with Trypsin/EDTA-solution for counting and passaging.

Serial dilutions with different cell concentrations were performed in 50 ml cubes with, e.g., 50,000 cells in 5 ml medium (corresponding to 200 cells in 20 µl). Further dilutions 1:2, 1:4, and 1:8 (=25 cells in 20 µl) were performed.

**Method of hanging droplets**. A black cross mark on the lid of a 6 cm culture plate separates four different regions. Several 20 µL droplets with different cell concentrations are pipetted on the lid, which is then carefully turned upside down to be placed on the lower part of the culture plate. 2.5 mL phosphate buffer are added to the plate for sufficient humidity in the local atmosphere and to prevent medium up-concentration. Cells are cultivated at 37 °C the $CO_2$ concentration of 5% until the cell clusters inside the hanging droplets have grown to the desired size.

**Reporting summary**. Further information on research design is available in the Nature Research Reporting Summary linked to this article.

## Data availability

Data supporting the main figures is available from the corresponding authors only upon reasonable request. Further information regarding design and cell preparations may be found in the Nature Research Reporting Summary.

## Code availability

Analysis codes are available from the corresponding author only upon reasonable request.

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

## Acknowledgements

The authors thank Birgit Erhard for cell preparation, Richard Bowman for help with hologram computations as well as Natalia Schatz and Oswald Prucker for coating of coverslips. We further thank Matthias Koch, Felix Jünger, Rebecca Michiels, and Philipp v. Olshausen for a thorough reading of the manuscript. This study was supported by the Excellence Initiative of the German Federal and State Governments, grants EXC 294 (BL) and EXC 2189 (Y).

## Author contributions

A.R. initiated the project. B.L. and A.R. designed the setup. B.L. performed all experiments, B.L. and A.R. developed the theory, Y performed BPM simulations, B.L. and A.R. analyzed data, and prepared figures. A.R. wrote the manuscript.

## Funding

## Competing interests

The authors declare no competing interests.
