## [Peer Review File · Nature Communications]

Reviewers' Comments:

Reviewer #1 (Remarks to the Author):

Review of: "Towards non-blind optical tweezing: Finding multiple grabbing positions from off-focus beam deformations"

by Benjamin Landenberger & Alexander Rohrbach

In the introduction, this manuscript offers the tantalising prospect of a major advance in optical tweezing: the ability to precisely control the 3D position and orientation of complicated mesoscopic objects of inhomogeneous spatially refractive index. Such a capability would offer new 3D tomographic imaging methods. It is undoubtedly a challenging problem. The introduction and the motivation for the work are well written, but perhaps this goal is over-emphasised (e.g. the first 2 figures are devoted to explaining it), given that the rest of the paper then only takes some steps towards achieving this challenging goal.

Equation 1 in the manuscript tells us that in order to impart significant optical forces to the sample, high intensity gradients (such as those found in focussed beams) must be overlapped with regions of high spatial changes in refractive index. Therefore it is important to identify how the refractive index changes throughout the sample.

To pursue this goal, the authors then describe what I understand to be a conceptually novel method (Off-Focus Interferometric Tracking -OFI) to attempt to map the 3D refractive index variations within an inhomogeneous sample such as a clump of cells. Describing this method, and some initial experiments using it, then forms the bulk of the rest of the paper. Unfortunately, the reader does not find results showing the application of OFI to control the motion of a free-floating sample using optical tweezers, which was discussed in depth in the introduction.

I think the OFI method itself is interesting, and in my view complementary to existing methods to probe similar information about a sample, such as phase contrast microscopy and refractive index tomography. However, given OFI shares its roots with some of these other techniques, the current manuscript does not give a clear enough account of how OFI compares in terms of accuracy, and time taken to recover the information. The theory of OFI is well introduced and clearly annotated in Fig 3. However the theory does contain some relatively severe approximations (mainly that we are in the single scattering regime). I think it would help if these approximations were more clearly highlighted - especially as we need to know if they cause OFI to fail under the more complicated scenarios it is used in later in the manuscript.

OFI is first demonstrated on a very simple sample - one in which the refractive index is essentially uniform, punctuated sparsely by spherical particles of a higher refractive index. A convincing experimental demonstration of OFI is given in this case in Fig 4.

OFI is next applied to samples of more complicated three dimensional structure. The results in this case look impressive. However it is difficult to validate their accuracy. The main concern I have here is that when OFI is applied within a sample of light emanating away from the focus to the imaging plane is subject to more diffraction and scattering from other parts of the sample. Given that the technique is interferometric, and the signals require determining the centre-of-mass of the resulting interference patterns, it is unclear to me how robust OFI is to diffraction from spatially varying refractive index around the point of interest.

To make a more convincing case here it would have been good to see the technique in action addressing the challenge of 'non-blind' optical trapping, for example by showing how the stiffness of an extended optically trapped object is increased by placing trapping beams in locations governed by OFI measurements. Or at least it might be useful to provide a simple simulation demonstrating that OFI does indeed work inside three dimensionally inhomogeneous samples. For example, the beam propagation method (e.g. Mansuripur, Masud, Ewan M. Wright, and Mahmoud Fallahi. "The beam propagation method." Optics and Photonics News 11.7 (2000): 42-48.) would

be a simple way to provide supporting evidence that OFI is robust even when the focus is formed in an inhomogeneous environment.

Given the above, while I do think the manuscript is interesting and the authors should be congratulated for tackling an important and difficult challenge, I do not think the manuscript provides sufficient experimental demonstrations beyond the state-of-the-art in sample driven holographic beam shaping at this stage (e.g. see Kim, Kyoohyun, and YongKeun Park. "Tomographic active optical trapping of arbitrarily shaped objects by exploiting 3D refractive index maps." *Nature communications* 8.1 (2017): 1-8.). Therefore unless the manuscript is extended to include a demonstration of OFI improving the trapping stiffness of an extended optically trapped object, then I recommend publication in a more technical journal.

Other smaller points:

Page 2 line 50: formatting, start of sentence

Equation 1 contains an approximations. Please specify what are these approximations? I assume it ignores radiation pressure - anything else? Please explain why these approximations are valid in all envisaged uses of OFI .

Figure 1: Please mark the rotation axis on the snapshots - this would help greatly to visualise what is happening.

page 4 line 137: 170um large? Please make the scale of the cell clearer: what is the approximate radius/diameter of the cell, is it spherical?

page 5, line 168: Please add an explanation or the appearance of the $\pi/2$ phase shift in the scattered light. What assumptions are made about the scattering material for this term to be $\pi/2$? Does this imply that there is no absorption? If there is absorption does this change the $\pi/2$ constant in a predictable way?

page 6, line 200: please add more detail about the methods used in the rigorous calculation, and the paraxial approximation of $I_{i,s}$, so that readers may be able to repeat your work if they wish.

Page 10 line 318, and page 12, line 430: Some symbols in this equation appeared as white filled boxes (I think they are all approximately equal to symbols?).

Page 11 line 375: typo 'fobtained' should be 'obtained'.

Reviewer #2 (Remarks to the Author):

The manuscript entitled "Towards non-blind optical tweezing: Finding multiple grabbing positions from off-focus beam deformations" by authors B. Landenberger and A. Rohrbach presents a technique to find large variations in the refractive index within a biological sample in order to optimize manipulation with holographic optical tweezers (HOTs).

As opposed to the "blind" alternative that consist in placing the optical traps randomly over the specimen, the authors argue that their targeted, "non-blind" optical manipulation approach offers improved performance for manipulation and that this can be applied with an advantage to optical microscopy.

The paper, except for a couple of editing errors, is well written, understandable, with nice figures and technically well done.

Furthermore, I find very interesting the fact that they can derive usable interferometric signals in a region so close to the imaging plane, where the intensity distributions are still spatially limited so

that these do not significantly overlap with those of neighbouring traps. Signal overlap at the back focal plane for multiple, permanent optical traps has impeded finding more quantitative applications for HOTs since their inception.

However, unfortunately, despite these positive features, I remain rather sceptical about the motivation and impact of the manuscript. I do not think the authors do a good job in convincing the reader of the significance of their optical manipulation technique for the microscopic observation of large biological specimens. Without a proper rationale, the interest of the manuscript is naturally much lower and thus, in my opinion, inappropriate for Nat. Commun. In the introduction they do not really explain why and when the observation of samples from multiple viewpoints may be advantageous in microscopy. Introductory section "Life-science microscopy" is too short, generic and contains no meaningful references. Considering the multiplicity of optical sectioning microscopies (confocal, light sheet, multiphoton) and clearing reagents for highly scattering samples currently available for 3D imaging in biology, this section appears naive.

Furthermore, it is easy to find prior work on alternative techniques for sample manipulation that the authors have missed. For example:

Berndt, F., Shah, G., Power, R.M. et al. "Dynamic and non-contact 3D sample rotation for microscopy," Nat Commun 9, 5025 (2018).

This is unfortunate because Berndt et al. paper does mention optical manipulation methods and argues against these. Berndt et al. also embed magnetic beads on their samples, which raises the question of why embedding dielectric beads is not discussed in the manuscript, since these can provide an improved grip over natural structures and resorting to artificial handles is absolutely acceptable in most experiments involving optical trapping, including manipulation within cells. Another aspect that contributes to make the paper unconvincing for the purported application is their use of bright-field microscopy, which is rarely employed for visualization of such complex samples. If bright-field is not a pre-requisite, cell constituents of high refractive index (such as membrane lipids or protein concentrations) can be labelled with fluorescent probes for easy identification under confocal microscopy, for example.

A final criticism on conceptual matters is the lack of quantitative information regarding the performance improvement relative to "blind" manipulation. I understand that this is difficult to obtain and sample-dependent but the anecdotal evidence provided by the manuscript is insufficient to claim that the method represents "a big step towards fully computer-controlled orientation and feature-optimized laser scanning of sub-mm sized biological specimens", as they do.

Regarding more technical issues, I should point out that I feel rather uncomfortable with the approximate reasoning the authors use throughout the manuscript. Very particularly, when they arrive at Eq. 6, which states that signals, index gradients and optical forces are proportional, I fail to see why the proportionality factors remain fixed regardless of the sample point.

In my opinion, similarly to what happens with the traditional stiffness calibration, these factors are not constant when they move through an inhomogeneous sample so that their index maps do not have a constant scale and cannot provide accurate comparative information between points.

A final issue the authors should consider worth clarifying is the tracking of the centroids of the intensity distributions. I understand that they are using the nominal values computed from the trap positions and magnification relationships between the different planes. I wonder how the strong scattering and optical aberrations produced within most 3D cell cultures can affect the computation of the OFI signals.

In summary, I think this is a paper with some merits but still in a proof-of-concept stage that raises more questions than answers. At this point, it appears to me as a solution in search of a problem. I advise the authors to send the manuscript to an optics journal such as Optics Express, in which they surely will have no trouble publishing, and try again a higher visibility journal if they manage to sufficiently enhance the concept at a later stage.

Reviewer #3 (Remarks to the Author):

Review of "Towards non-blind optical tweezing: Finding multiple grabbing positions from off-focus beam deformations" by Benjamin Landenberger and Alexander Rohrbach.

In this manuscript, the authors present a method to localize 3D grabbing positions for holographic optical traps towards non-blind optical tweezing, by off-focus interferometric (OFI) imaging of refractive index changes of the cell clusters. The basic principle governing this method is that the optical position is characterized by sufficiently strong optical gradient forces, which depend on local intensity gradient and spatial change in refractive index. That means if one can map the 3D refractive index of the specimen, the non-blind optical trapping can be achieved by generating the desired local intensity gradient.

In this work, the authors adopt OSI imaging to track regions with increased refractive index and then use the information to further estimate the grabbing positions for optical traps. However, there is already a very similar work [Nat Commun 8, 15340 (2017).] that has achieved optical control of arbitrarily shaped particles including orientation of cells. More specifically, that work actually employs optical diffraction tomography (ODT) to precisely map 3D refractive index distribution of samples in real time, and then generate the 3D light mould for stable trapping. In other words, these two works share the same principle, but adopt different imaging methods to obtain the refractive index information. From this point of view, I do not see obvious advantages of the present work over the published one, although this work presents very good concept demonstration and results.

By the way, this important work is not mentioned in the current manuscript. The authors might miss this reference. Therefore, I suggest the authors clarify the novelty of this work based on adequate discussion of related work before resubmitting it for further review.

Dear reviewers,

thanks a lot for the time you spent to carefully read our manuscript and for your comments, which helped to improve our study. I'm sorry for the significant time delay in my response, but I had to introduce a new PhD student how to do generate further OFI data.

Kind regards,

Alexander Rohrbach

Reviewers' comments:

Reviewer #1 (Remarks to the Author):

Review of: "Towards non-blind optical tweezing: Finding multiple grabbing positions from off-focus beam deformations"

by Benjamin Landenberger & Alexander Rohrbach

In the introduction, this manuscript offers the tantalising prospect of a major advance in optical tweezing: the ability to precisely control the 3D position and orientation of complicated mesoscopic objects of inhomogeneous spatially refractive index. Such a capability would offer new 3D tomographic imaging methods. It is undoubtedly a challenging problem. The introduction and the motivation for the work are well written, but perhaps this goal is over-emphasised (e.g. the first 2 figures are devoted to explaining it), given that the rest of the paper then only takes some steps towards achieving this challenging goal.

Reply: It is true that we spent a substantial amount of the paper to motivate our goal. Other reviewers before wanted to have clarified what is really new with this study. If researchers have not worked a long time on optical tweezers and all the developments in the field, one may easily come to the wrong conclusion that our study does not present a new method. I am relieved that you see the challenging problem. However, the solution to this also works without 3D tomography (in the classical sense).

Equation 1 in the manuscript tells us that in order to impart significant optical forces to the sample, high intensity gradients (such as those found in focussed beams) must be overlapped with regions of high spatial changes in refractive index. Therefore it is important to identify how the refractive index changes throughout the sample.

Other smaller points:

Equation 1 contains an approximations. Please specify what are these approximations? I assume it ignores radiation pressure - anything else? Please explain why these approximations are valid in all envisaged uses of OFI .

Reply: It is correct that we left out the radiation pressure, since it is manifold smaller than the gradient forces especially at high focusing angles of NA = 1.2. Furthermore, by inserting the NIR-trapping laser from the bottom, the rotation pressure is nearly compensated by the gravity of the cell clusters. In addition, we used an approximation for the Clausius Mossotti polarizability

$$\alpha(\mathbf{b}) = 3V \cdot \frac{(n(\mathbf{b})+\Delta n(\mathbf{b}))^2 - n(\mathbf{b})^2}{(n(\mathbf{b})+\Delta n(\mathbf{b}))^2 + 2n(\mathbf{b})^2} \stackrel{\Delta n^2 \ll n^2}{\approx} 3V \cdot \frac{(n^2 + 2n\Delta n) - n^2}{(n^2 + 2n\Delta n) + 2n^2} = V \cdot \frac{2\Delta n(\mathbf{b})}{n(\mathbf{b})} \text{ for } 0.15^2 = \Delta n^2 \ll n^2 = 1.38^2$$

which is reasonable for refractive index changes $\Delta n < 0.15$.

Therefore $\mathbf{F}_{\text{opt}}(\mathbf{b}) \approx \mathbf{F}_{\text{grad}}(\mathbf{b}) = \frac{n(\mathbf{b})}{2c} \cdot \alpha(\mathbf{b}) \cdot \nabla I_1(\mathbf{b}) \approx \frac{V}{c} \cdot \Delta n(\mathbf{b}) \cdot \nabla I_1(\mathbf{b})$

We added this extra explanation to equation 1.

To pursue this goal, the authors then describe what I understand to be a conceptually novel method (Off-Focus Interferometric Tracking -OFI) to attempt to map the 3D refractive index variations within an inhomogeneous sample such as a clump of cells. Describing this method, and some initial experiments using it, then forms the bulk of the rest of the paper. Unfortunately, the reader does not find results showing the application of OFI to control the motion of a free-floating sample using optical tweezers, which was discussed in depth in the introduction.

Reply: yes, but this is undoubtedly the challenging problem, how you phrased it in your first paragraph. It will require quite a while to achieve this goal in a satisfactory manner. Here we show the concept, which is adventurous and novel enough - in my opinion.

I think the OFI method itself is interesting, and in my view complementary to existing methods to probe similar information about a sample, such as phase contrast microscopy and refractive index tomography. However, given OFI shares its roots with some of these other techniques, the current manuscript does not give a clear enough account of how OFI compares in terms of accuracy, and time taken to recover the information. The theory of OFI is well introduced and clearly annotated in Fig 3. However the theory does contain some relatively severe approximations (mainly that we are in the single scattering regime). I think it would help if these approximations were more clearly highlighted - especially as we need to know if they cause OFI to fail under the more complicated scenarios it is used in later in the manuscript.

Reply: Yes, OFI does share roots with the above mentioned methods, since it bases on contrast formation from refractive index changes. And refractive index changes are the basis to apply optical forces. However, as pointed out in the manuscript, we do not need any visible light for imaging, since the optical traps, i.e. the laser foci are supposed to find the local refractive index changes "by themselves" (i.e. by sensing and moving the fingers as I try to illustrate in the last figure with the potato). The best position (with maximum refractive index change in the focus) can be found by analyzing the forward scattered light. And, we do not have to perform any reconstructions, besides the simple image analysis to calculate $S_x(\mathbf{b})$, $S_y(\mathbf{b})$ or $S_{\perp}(\mathbf{b})$.

Another approximation, i.e. that $\left| \sum_{n=1}^N (E_{i,n} + E_{s,n}(\mathbf{b}_n)) \right|^2 \approx \sum_{n=1}^N |E_{i,n} + E_{s,n}(\mathbf{b}_n)|^2$ means that the diffraction patterns at the camera do not overlap. This can be achieved reasonably well for $N < 10$ particles and an adequate camera position at z_d , as shown in figure 4b. I added short sentence above equation (2).

Since Nature Communications is an interdisciplinary Journal, I didn't want to spend too much emphasis on complex formulas, especially since no reconstructions are necessary. So, eqs. (1) and (2) are simplified descriptions, but put the problems with the best grabbing positions and the diffraction patterns in a nutshell.

But I agree with you: there is a big step between single particle scattering and multiple scattering and I fully accept your skepticism. So, we added another figure (Fig.5) and explanation to close this gap, hoping to make our results shown in the last figure more convincing.

OFI is first demonstrated on a very simple sample - one in which the refractive index is essentially uniform, punctuated sparsely by spherical particles of a higher refractive index. A convincing experimental demonstration of OFI is given in this case in Fig 4.

OFI is next applied to samples of more complicated three dimensional structure. The results in this

case look impressive. However it is difficult to validate their accuracy. The main concern I have here is that when OFI is applied within a sample of light emanating away from the focus to the imaging plane is subject to more diffraction and scattering from other parts of the sample. Given that the technique is interferometric, and the signals require determining the centre-of-mass of the resulting interference patterns, it is unclear to me how robust OFI is to diffraction from spatially varying refractive index around the point of interest.

Reply: Absolutely, I see your point. And before this study I have had the same opinion as you. So we generated a new figure to make this point clearer.

To make a more convincing case here it would have been good to see the technique in action addressing the challenge of 'non-blind' optical trapping, for example by showing how the stiffness of an extended optically trapped object is increased by placing trapping beams in locations governed by OFI measurements. Or at least it might be useful to provide a simple simulation demonstrating that OFI does indeed work inside three dimensionally inhomogeneous samples. For example, the beam propagation method (e.g. Mansuripur, Masud, Ewan M. Wright, and Mahmoud Fallahi. "The beam propagation method." *Optics and Photonics News* 11.7 (2000): 42-48.) would be a simple way to provide supporting evidence that OFI is robust even when the focus is formed in an inhomogeneous environment.

Reply: Since first author Benjamin Landenberger did all the experiments and has left my group a while ago, it was impossible to perform further experiments for this manuscript. So, I went the other way that you proposed: A recently hired PhD student (new second author Yatish) performed thousands of beam propagation method (BPM) simulations to prove that the refractive index changes of individual spheres (1.1 μm in diameter) can be seen even in a 50 μm cluster consisting of nearly 5000 spheres. I find the result amazing and convincing, keeping in mind that each wavefront of the coherent beam is heavily distorted by the other particles before and behind the sphere of interest. OFI imaging is a partially coherent technique, since it consists of coherently scattered light, where each diffraction pattern (analyzed with the described procedure) is added incoherently to the next one through the scanning process. For details please find the new figure 5 and the corresponding descriptions and discussions. The new figure tells us that the most prominent phase perturbations occur by the structures in the center of the focus, enabling OFI coherent imaging/tracking even through strongly scattering media.

Given the above, while I do think the manuscript is interesting and the authors should be congratulated for tackling an important and difficult challenge, I do not think the manuscript provides sufficient experimental demonstrations beyond the state-of-the-art in sample driven holographic beam shaping at this stage (e.g. see Kim, Kyoohyun, and YongKeun Park. "Tomographic active optical trapping of arbitrarily shaped objects by exploiting 3D refractive index maps." *Nature communications* 8.1 (2017): 1-8.). Therefore unless the manuscript is extended to include a demonstration of OFI improving the trapping stiffness of an extended optically trapped object, then I recommend publication in a more technical journal.

Reply: It was our mistake to overlook the publication from Kim & Park. This is really impressive work. However, it's totally different from our method. In the introduction, I discuss that Kim & Park rotated small objects such as single cells (and not cell clusters of hundreds or thousands of cells) by first performing a 3D tomographic scan to reconstruct the cells' 3D refractive index distributions. In a next step, the authors calculated and holographically generated a 3D coherent light distribution, which exerts an averaged optical torque onto the cell in different rotation

directions. However, because of their intensity gradients, which are manifold weaker than those of point traps, only small objects can be rotated. On the one hand this is nicely pointed out in their figure 1. On the other hand, I contacted both YongKeun Park and first author Kyoohyun Kim, to clarify this issue. I had extensive email exchange with YongKeun's postdoc Kyoohyun, who confirmed the profound differences in the two techniques and who allowed me to share the email discussion for this review process (see below).

In summary, we would be very glad, if reviewer #1 could change his mind based on the new advanced computer simulations we performed and based on the explanations given above, especially about the differences to the method of Kim & Park.

Other smaller points:

Page 2 line 50: formatting, start of sentence

Equation 1 contains an approximations. Please specify what are these approximations? I assume it ignores radiation pressure - anything else? Please explain why these approximations are valid in all envisaged uses of OFI .

Reply: see above

Figure 1: Please mark the rotation axis on the snapshots - this would help greatly to visualise what is happening.

Reply: we added a little sketch showing more clearly that rotation is around the x-axis

page 4 line 137: 170um large? Please make the scale of the cell clearer: what is the approximate radius/diameter of the cell, is it spherical?

Reply: as written directly above figure 2 the cell cluster has a diameter of 170 μm . It consists of about 2000 cells with a diameter of 10-15 μm . We added the cell size and number to the text.

page 5, line 168: Please add an explanation or the appearance of the $\pi/2$ phase shift in the scattered light. What assumptions are made about the scattering material for this term to be $\pi/2$? Does this imply that there is no absorption? If there is absorption does this change the $\pi/2$ constant in a predictable way?

Reply: The $\pi/2$ phase shift is only precise for a non-absorbing radial scatterer, having a higher refractive index than the environment. However, since our paper does not rely on any exact or numerical calculation, we want to make the concepts clear to the reader. Approximating a $\pi/2$ phase simplifies the mathematical description as pointed out in eq. (2). The approximation for the phase shift is nowhere important in our study.

page 6, line 200: please add more detail about the methods used in the rigorous calculation, and the paraxial approximation of $I_{\{i,s\}}$, so that readers may be able to repeat your work if they wish.

Reply: $I_{\{i,s\}}$ is the intensity difference, from this the signals $S_x(b)$, $S_y(b)$ and are calculated $S_z(b)$ by varying the beam position b . The Rayleigh scattering with the (paraxial) Gaussian beam could be saved in space domain. The rigorous calculation has been performed by a software ("LightWave") which I have written in C during my Post-Doc time at EMBL and performs many operations in k -space. It is based on a superposition of thousands of plane waves, forming an arbitrarily shaped incident beam (here highly focused beam), where each plane wave (having a different azimuthal and polar angle) scatters at either spheres (using Mie theory) or at e.g. rods (using the

Born/Rayleigh-Gans approximation). This requires heavy usage of Euler rotation matrices. Several of my papers published before 2010 were based on this method / software, which has been checked for correctness with other approaches provided by collaboration partners from other universities.

Page 10 line 318, and page 12, line 430: Some symbols in this equation appeared as white filled boxes (I think they are all approximately equal to symbols?).

Reply: sorry for this. Yes, these are exactly the symbols, which are well read readable in my document.

Page 11 line 375: typo 'fobtained' should be 'obtained'. Okay, thanks.

Reviewer #2 (Remarks to the Author):

The manuscript entitled "Towards non-blind optical tweezing: Finding multiple grabbing positions from off-focus beam deformations" by authors B. Landenberger and A. Rohrbach presents a technique to find large variations in the refractive index within a biological sample in order to optimize manipulation with holographic optical tweezers (HOTs).

As opposed to the "blind" alternative that consist in placing the optical traps randomly over the specimen, the authors argue that their targeted, "non-blind" optical manipulation approach offers improved performance for manipulation and that this can be applied with an advantage to optical microscopy.

The paper, except for a couple of editing errors, is well written, understandable, with nice figures and technically well done.

Furthermore, I find very interesting the fact that they can derive usable interferometric signals in a region so close to the imaging plane, where the intensity distributions are still spatially limited so that these do not significantly overlap with those of neighbouring traps. Signal overlap at the back focal plane for multiple, permanent optical traps has impeded finding more quantitative applications for HOTs since their inception.

However, unfortunately, despite these positive features, I remain rather sceptical about the motivation and impact of the manuscript. I do not think the authors do a good job in convincing the reader of the significance of their optical manipulation technique for the microscopic observation of large biological specimens. Without a proper rationale, the interest of the manuscript is naturally much lower and thus, in my opinion, inappropriate for Nat. Commun. In the introduction they do not really explain why and when the observation of samples from multiple viewpoints may be advantageous in microscopy. Introductory section "Life-science microscopy" is too short, generic and contains no meaningful references. Considering the multiplicity of optical sectioning microscopies (confocal, light sheet, multiphoton) and clearing reagents for highly scattering samples currently available for 3D imaging in biology, this section appears naive.

Reply: It is true that one can argue about the most efficient, popular and best applicable methods that will be used to hold extended spacemen in the next 10 to 20 years. We and many others (including the company Carl-Zeiss) think that rotation of objects (< 0.5mm) by optical forces is a rather promising, yet challenging approach. Without doubt the orientation and position of the object plays a decisive role, whether certain features can be recovered or not. Sure, this does not hold true for all problems and biological questions.

However, we agree that the introduction about life science microscopy (in thick media) could be improved. So we rephrased this section and added several references, giving a better motivation for imaging applications.

Furthermore, it is easy to find prior work on alternative techniques for sample manipulation that the authors have missed. For example:

Berndt, F., Shah, G., Power, R.M. et al. "Dynamic and non-contact 3D sample rotation for microscopy," *Nat Commun* 9, 5025 (2018).

This is unfortunate because Berndt et al. paper does mention optical manipulation methods and argues against these. Berndt et al. also embed magnetic beads on their samples, which raises the question of why embedding dielectric beads is not discussed in the manuscript, since these can provide an improved grip over natural structures and resorting to artificial handles is absolutely acceptable in most experiments involving optical trapping, including manipulation within cells.

Reply: I cannot remember why we did not cite this paper. I agree with you that it should be mentioned in the introduction, since it represents another contactless method to rotate or re-position objects. So, in the revised manuscript version we mention this paper explicitly at the beginning of the introduction, since it has nothing to do with optical tweezers. Especially not with our novel approach of non-blind optical tweezing. It is known and of course I agree, that magnetic forces can be significantly larger than optical forces, making the approach from Berndt especially useful for objects with sizes of 1 mm and larger. However, I do not share his argument against optical force based manipulation.

Another aspect that contributes to make the paper unconvincing for the purported application is their use of bright-field microscopy, which is rarely employed for visualization of such complex samples. If bright-field is not a pre-requisite, cell constituents of high refractive index (such as membrane lipids or protein concentrations) can be labelled with fluorescent probes for easy identification under confocal microscopy, for example.

Reply: I think our method is compelling since it does not require any additional handles such as magnetic beads and it does not require any fluorescence labeling to make high refractive index samples visible. As discussed with reviewer #1, we do not require brightfield imaging at all, since the NIR-laser tweezers "fingers" are supposed to sense the local refractive index changes, by reading out the forward scattered light. I think this is a rather new approach and it has been rather unexpected that these refractive index changes can be detected in such inhomogeneous media. However, I have declared from the beginning that this is a concept study. Also, theory papers reveal concepts and are per-se unable to provide experimental (!) verification.

A final criticism on conceptual matters is the lack of quantitative information regarding the performance improvement relative to "blind" manipulation. I understand that this is difficult to obtain and sample-dependent but the anecdotal evidence provided by the manuscript is insufficient to claim that the method represents "a big step towards fully computer-controlled orientation and feature-optimized laser scanning of sub-mm sized biological specimens", as they do.

Reply: After having translated "anecdotal evidence" I realize that you're not happy with our results. Since this is a concept study, it's impossible to demonstrate any performance improvement. Definitely this must be achieved in follow-up papers over the next years. In this

paper we demonstrate 4 things: i) we show that optical trapping is usually blind and that nonblind optical trapping is possible. ii) We demonstrate that optical forces are strong enough to rotate large objects around the optical axis, but not around the X or Y axis. iii) based on the knowledge that optical forces are proportional to the refractive index change, we show that strong(est) refractive index changes (RCI) can be found for each optical trap individually. iv) we show that the RCI can be found without imaging, but purely by reading out the scattered light.

Regarding more technical issues, I should point out that I feel rather uncomfortable with the approximate reasoning the authors use throughout the manuscript. Very particularly, when they arrive at Eq. 6, which states that signals, index gradients and optical forces are proportional, I fail to see why the proportionality factors remain fixed regardless of the sample point. In my opinion, similarly to what happens with the traditional stiffness calibration, these factors are not constant when they move through an inhomogeneous sample so that their index maps do not have a constant scale and cannot provide accurate comparative information between points.

Reply: I understand your concern and this is why we provided further evidence. As reviewer #1 pointed out correctly, the step from tracking single particles (= a local RCI) in a homogeneous environment relative to finding a local RCI inside the cluster of thousands of particles has been too big. Therefore, we performed intensive computer simulations with the beam propagation method (BPM), to show that detecting different RCIs is possible despite strong wavefront perturbations of the coherent laser light. This important, and hopefully convincing result is summarized in the new figure 5. We do not have to measure the forces by calibrations or finding stiffnesses, we simply have to find the best local grabbing position from the strongest refractive index change.

A final issue the authors should consider worth clarifying is the tracking of the centroids of the intensity distributions. I understand that they are using the nominal values computed from the trap positions and magnification relationships between the different planes. I wonder how the strong scattering and optical aberrations produced within most 3D cell cultures can affect the computation of the OFI signals.

Reply: Yes, we are using the nominal values. As mentioned above the multiple scattering affects the RCI information transfer from the focus center (the grabbing point) to the detector much less than most people expected - me included some years ago.

In summary, I think this is a paper with some merits but still in a proof-of-concept stage that raises more questions than answers. At this point, it appears to me as a solution in search of a problem. I advise the authors to send the manuscript to an optics journal such as Optics Express, in which they surely will have no trouble publishing, and try again a higher visibility journal if they manage to sufficiently enhance the concept at a later stage.

Reply: I hope that I could answer most of your questions, both with my reply to the comments and with the additional computer simulations. I deeply regret that you consider our study as a solution in search of a problem.

Reviewer #3 (Remarks to the Author):

Review of "Towards non-blind optical tweezing: Finding multiple grabbing positions from off-focus

beam deformations” by Benjamin Landenberger and Alexander Rohrbach.

In this manuscript, the authors present a method to localize 3D grabbing positions for holographic optical traps towards non-blind optical tweezing, by off-focus interferometric (OFI) imaging of refractive index changes of the cell clusters. The basic principle governing this method is that the optical position is characterized by sufficiently strong optical gradient forces, which depend on local intensity gradient and spatial change in refractive index. That means if one can map the 3D refractive index of the specimen, the non-blind optical trapping can be achieved by generating the desired local intensity gradient.

In this work, the authors adopt OSI imaging to track regions with increased refractive index and then use the information to further estimate the grabbing positions for optical traps. However, there is already a very similar work [Nat Commun 8, 15340 (2017).] that has achieved optical control of arbitrarily shaped particles including orientation of cells. More specifically, that work actually employs optical diffraction tomography (ODT) to precisely map 3D refractive index distribution of samples in real time, and then generate the 3D light mould for stable trapping. In other words, these two works share the same principle, but adopt different imaging methods to obtain the refractive index information. From this point of view, I do not see obvious advantages of the present work over the published one, although this work presents very good concept demonstration and results.

Reply: I totally agree with you that we should have cited this paper. I simply missed this strong paper. However, as explained already to referee #1, the technique from Kim and Park is quite different: as discussed with first author Kim, he cannot rotate large objects consisting of hundreds of thousands of cells, as we can do it by blind optical trapping and rotation only around the z axis. Since we use highly focused point traps, our gradient forces are significantly stronger than their holographically shaped intensity distribution. Furthermore, they use optical diffraction tomography (ODT) to obtain a global refractive index map, whereas we do not require imaging at all, since the multiple optical traps reveal their good or bad (high or low RCI) position purely by reading out the scattered light in the local environment. Hence, the range of applications is different, the optical trapping is quite different and the detection of refractive index changes (RCI) is quite different.

By the way, this important work is not mentioned in the current manuscript. The authors might miss this reference. Therefore, I suggest the authors clarify the novelty of this work based on adequate discussion of related work before resubmit it for further review.

Reply: We added a new figure with new results to our manuscript (to improve the line of arguing), and we added a brief discussion about the above mentioned related papers that we missed

Last email to me from first author Kyoohyun Kim of the paper:

Kyoohyun Kim, and YongKeun Park. "Tomographic active optical trapping of arbitrarily shaped objects by exploiting 3D refractive index maps." Nature communications 8.1 (2017): 1-8.)

He confirmed the differences of our concepts, which I described above.

REVIEWER COMMENTS

Reviewer #1 (Remarks to the Author):

2nd round review of:

“Towards non-blind optical tweezing: Finding multiple grabbing positions from off-focus beam deformations”

by Benjamin Landenberger, Yatish, & Alexander Rohrbach

The authors have considered my previous comments carefully. In my view the addition of Figure 5, a simulation of the performance of OFI in the middle of a cluster of 4900 1.1 μ m sized spheres improves the readers confidence in the technique.

There are some points that are unclear to me about Fig. 5:

- In 5a, the off-focus camera signals are difficult to see clearly - I suggest to enlarge and/or make into 2D panels.
- In 5b, what does the greyscale value of the spheres in the cutaway represent? i.e. some are white, some are black and some are grey - does this have any significance or is it just to show the sphere locations? Please add a scale if it means something.
- In 5b what is the significance of the grey background disk? Are the spheres embedded in a single large sphere of higher refractive index than water or is this just to indicate their boundary? Please explain.
- It is a little difficult to compare the OFI results with the actual locations of the spheres to see how accurate it is. Please place extra panels adjacent to each OFI image showing the true location of each sphere across the scanned planes.
- Why does the yellow panel have higher intensities at the top and bottom, and lower intensities across the middle? Why does the blue panel have higher intensities in the bottom left corner where the beam focus should be outside the cluster of spheres? Why do the red and yellow panels have lower intensities in the middle and higher at the edge? Are these systematic errors - please explain how these effects arise, and what they mean for the validity of the technique.

Finally, given that the manuscript only goes as far as demonstrating OFI, and not their envisaged future application of non-blind optical tweezing, Perhaps they will consider changing the title to reflect this, by focussing on the term 'Off-focus interferometric tracking' and adding a paragraph discussing what other applications OFI may have beyond optical tweezing.

In summary, I still have some reservations about the way the manuscript is presented - I should like to see more focus in the introduction on the OFI technique itself rather than the non-blind optical tweezing application (which in my view are still not demonstrated strongly in

the paper). However, on balance I think OFI is an interesting new technique, and I am now more confident in its applicability to samples of inhomogeneous refractive index, and so I now recommend publication in Nature Communications once the above points have been answered.

Reviewer #2 (Remarks to the Author):

Second review to "Towards non-blind optical tweezing: Finding multiple grabbing positions from off-focus beam deformations" by B. Landenberger, Yatish and A. Rohrbach

I have read the author's reply and revised the improved manuscript and, while I value the new content and especially the new simulation, I think the authors do not address sufficiently well my key concerns, which I list below again, so that I cannot recommend this manuscript for publication in Nat. Commun.

1) Relevance to microscopy of thick samples.

The authors have improved the introduction somewhat but when it comes to justifying the need for sample rotation they simply state: "observation from different directions is the only way out", with no references or specific discussion.

For example, rotation is not the only way to acquire multiple views in microscopy. Proven, commercially available techniques such as MuVi-SPIM solve the problem differently. Thus, a much more sophisticated analysis is needed if the authors want to convince the reader of the importance of their work.

However, they keep writing: "Non-blind optical trapping with point tweezers is still a vision, but is of utmost interest in modern 3D microscopy". Based on the information provided in the manuscript, this sentence appears as an exaggerated and unsubstantiated statement.

2) Anecdotal evidence.

Prof. Rohrbach writes in his reply: "it's impossible to demonstrate any performance improvement. Definitely this must be achieved in follow-up papers over the next years".

I do not agree. They should provide this evidence here. Considering the small refractive index changes observed in biological samples, this method may end up being only marginally better than the "blind" approach and thus of little use.

3) I fail to see why the proportionality factors remain fixed regardless of the sample point at Eq. 7.

My criticism is that the proportionality factor between signal and force in equation 7 (revised manuscript) changes as we move throughout the sample so that you cannot define a global force scale for comparison between different sample points.

I see not author reply regarding this concern.

In summary, I still think that this work is in a too speculative stage and the authors cannot convincingly argue that it will make any impact in microscopy as it stands.

Reviewer #3 (Remarks to the Author):

The revised manuscript is well improved with some new simulations and explanations. Also, the authors claim the differences between their work and that from Kim & Park. Indeed, this technique could be used to trap larger objects with multiple highly focused point traps. It is obvious, because the focused laser spots have stronger gradient forces. Actually, the two methods share the same roots. Unfortunately, we don't see much new physics from this point.

I agree with Reviewer #1 that this work is interesting and useful, but it might be better to publish in a more technical journal instead of Nature Communications.

Dear reviewers, dear editor,
thanks a lot for your time and your energy you put into our manuscript. Your critical comments and questions helped to improve the manuscript further. Please find below our answers to all of your questions and comments.

Kind regards,
Alexander Rohrbach

REVIEWER COMMENTS

Reviewer #1 (Remarks to the Author):

The authors have considered my previous comments carefully. In my view the addition of Figure 5, a simulation of the performance of OFI in the middle of a cluster of 4900 1.1 μ m sized spheres improves the readers confidence in the technique.

There are some points that are unclear to me about Fig. 5:

- In 5a, the off-focus camera signals are difficult to see clearly - I suggest to enlarge and/or make into 2D panels.

Reply: We agree and we have improved the visibility of the 2D panels.

- In 5b, what does the greyscale value of the spheres in the cutaway represent? i.e. some are white, some are black and some are grey - does this have any significance or is it just to show the sphere locations? Please add a scale if it means something.

Reply: The gray value of the spheres is just the size of the sphere slices. We added a gray scale bar as you proposed.

- In 5b what is the significance of the grey background disk? Are the spheres embedded in a single large sphere of higher refractive index than water or is this just to indicate their boundary? Please explain.

Reply: The gray shaded circular area just indicates the cross-section of the sphere cluster. There is no other refractive index than that of water. We added a sentence at the end of the figure caption.

- It is a little difficult to compare the OFI results with the actual locations of the spheres to see how accurate it is. Please place extra panels adjacent to each OFI image showing the true location of each sphere across the scanned planes.

Reply: okay, we added in two of the four cross-sections yellow dots indicating the tracked center positions of the sphere slices. Now the localization of the spheres relative to the sphere OFI images can be better seen. The other two cross-sections were left as before to illustrate the shape of OFI images.

- Why does the yellow panel have higher intensities at the top and bottom, and lower intensities across the middle? Why does the blue panel have higher intensities in the bottom left corner where the beam focus should be outside the cluster of spheres? Why do the red and yellow panels have lower intensities in the middle and higher at the edge? Are these systematic errors - please explain how these effects arise, and what they mean for the validity of the technique.

Reply: These attenuation effects at the edges of the panel stem from the reduced overlap of beams scanning through a 3D volume. Taking only the center volume from a larger volume would have avoided this edge effects, but the computation time would have been too many days. This effect has nothing to do with the precision or validity of the technique.

Finally, given that the manuscript only goes as far as demonstrating OFI, and not their envisaged future application of non-blind optical tweezing, Perhaps they will consider changing the title to reflect this, by focussing on the term 'Off-focus interferometric tracking' and adding a paragraph discussing what other applications OFI may have beyond optical tweezing.

Reply: The first part of the title "Towards nonblind optical tweezing:" indicates that we have not reached this demanding goal. But it illustrates that most other tweezing has been "blind" so far. And after 30 years of blind optical tweezing, I think this is a very important novel concept, although two cited papers mention this superficially or indirectly. OFI scattering is one proposal to reach this difficult goal. But you convinced me and I changed the 2nd part of the title to: "Finding multiple grabbing positions from off-focus interferometric (OFI) tracking". As you proposed, we added another 3 sentences in the discussion motivating possible applications of OFI imaging, and in addition a remark to the conclusions.

I summary, I still have some reservations about the way the manuscript is presented - I should like to see more focus in the introduction on the OFI technique itself rather than the non-blind optical tweezing application (which in my view are still not demonstrated strongly in the paper). However, on balance I think OFI is an interesting new technique, and I am now more confident in its applicability to samples of inhomogeneous refractive index, and so I now recommend publication in Nature Communications once the above points have been answered.

Reply: We are happy that we could change your mind. We added another sentence in the introduction about the idea of the OFI technique and changed the subheading to "Localization of RI changes and OFI tracking". In lines 109 - 125 we had already discussed other interferometric tracking techniques related to OFI.

Reviewer #2 (Remarks to the Author):

Second review to "Towards non-blind optical tweezing: Finding multiple grabbing positions from off-focus beam deformations" by B. Landenberger, Yatish and A. Rohrbach

I have read the author's reply and revised the improved manuscript and, while I value the new content and especially the new simulation, I think the authors do not address sufficiently well my key concerns, which I list below again, so that I cannot recommend this manuscript for publication in Nat. Commun.

1) Relevance to microscopy of thick samples.

The authors have improved the introduction somewhat but when it comes to justifying the need for sample rotation they simply state: "observation from different directions is the only way out", with no references or specific discussion.

Reply: My complete sentence in the introduction was "*Without toxic clearing techniques [5], cell clusters, small plants or embryos with sizes from tens of μm to few mm are often so thick that*

illumination light and/or fluorescence light is absorbed or scattered so strongly [6] that observation from different directions is the only way out [7]."

You might know that scattering in large specimen in light sheet microscopy has been one of my top research topics over the last 15 years. The necessity of multiview observation is well accepted in the light sheet community. I added another reference ([7] Preibisch 2014), in which the authors confirm this statement in the first sentence of their introduction, giving further references. Furthermore, I added the word "**often the only way out**" to consider the few examples of large specimen where rotation is not helpful. The rest of my paragraph handles other papers only concentrating on object rotation, indicating the importance of this possibility.

For example, rotation is not the only way to acquire multiple views in microscopy. Proven, commercially available techniques such as MuVi-SPIM solve the problem differently. Thus, a much more sophisticated analysis is needed if the authors want to convince the reader of the importance of their work.

Reply: Of course, there are systems with more observation directions, but nevertheless it is not given that the object is oriented in the way it should be or if it changes its shape and orientation during the observation time period. This also would require object rotation.

However, they keep writing: "Non-blind optical trapping with point tweezers is still a vision, but is of utmost interest in modern 3D microscopy". Based on the information provided in the manuscript, this sentence appears as an exaggerated and unsubstantiated statement.

Reply: We do not share your opinion that our statement about non-blind optical tweezing is exaggerated and unsubstantiated. Optical tweezers have become a really strong technique over the last 30 years (see Nobel prize), but applications to objects of tens of μm are very limited. Why? Because blind optical trapping fails too often. However, I exchanged the word "utmost" by "great".

2) Anecdotal evidence.

Prof. Rohrbach writes in his reply: "it's impossible to demonstrate any performance improvement. Definitely this must be achieved in follow-up papers over the next years".

I do not agree. They should provide this evidence here. Considering the small refractive index changes observed in biological samples, this method may end up being only marginally better than the "blind" approach and thus of little use.

Reply: A stable performance of non-blind optical trapping will require several years of research by several groups, for reasons we have explained in line 85 ff. However, we have shown in this manuscript that rotation of large objects also in the critical direction (rotation around an axis in the image plane) is possible, but that the laser foci slip repetitively relative to the object (see also SM video 3). Furthermore, refractive index changes are not small at the edges of the specimen, hence making them potential grabbing points for the optical traps.

3) I fail to see why the proportionality factors remain fixed regardless of the sample point at Eq. 7. My criticism is that the proportionality factor between signal and force in equation 7 (revised manuscript) changes as we move throughout the sample so that you cannot define a global force scale for comparison between different sample points.

I see not author reply regarding this concern.

Our Reply from the previous round: I understand your concern and this is why we provided further evidence. As reviewer #1 pointed out correctly, the step from tracking single particles (= a local refractive index change RIC) in a homogeneous environment relative to finding a local RIC inside the cluster of thousands of particles has been too big. Therefore, we performed intensive

computer simulations with the beam propagation method (BPM), to show that detecting different RICs is possible despite strong wavefront perturbations of the coherent laser light. This important, and hopefully convincing result is summarized in the new figure 5. We do not have to measure the forces by calibrations or finding stiffnesses, we simply have to find the best local grabbing position from the strongest refractive index change.

Additional reply: I'm still not 100% sure what you mean with fixed proportionality factors, but to clarify things, we slightly revised eq. (7) once more to show that both the OFI signal and the force change approximately linear with a change in the refractive index $\delta n(\mathbf{b})$, assuming an unchanged

focus volume: $\mathbf{S}_{\perp}(\mathbf{b}) \sim \iint \left(\frac{x}{y}\right) I_d(x, y, \delta n(\mathbf{b})) dx dy \sim \nabla n(\mathbf{b}) \sim \mathbf{F}_{opt}(\mathbf{b})$

Eq. (7) was and is a summary of eqs. (1),(3),(6), where we show in eq. (1) that the optical force $\mathbf{F}_{opt}(\mathbf{b}) \approx \frac{v}{c} \cdot \langle \Delta n(\mathbf{b}) \cdot \nabla I_1(\mathbf{b}) \rangle$ is approximately proportional to the refractive index change $\Delta n(\mathbf{b})$. In eq. (3), where we inserted a second line for clarification

$$\begin{aligned} S_{xn}(\mathbf{b}, z_{dn}) &= \frac{1}{S_n(\mathbf{b}, z_{dn})} \cdot \frac{1}{z_{dn}} \int_{\Omega_n} x \cdot I_{dn}(x, y, z_{d,n}, \mathbf{b}) dx dy \\ &\approx \frac{1}{S_n(\mathbf{b}, z_{dn})} \cdot \frac{1}{z_{dn}} \int_{\Omega_n} x \cdot I_{cn}(\mathbf{b}) \cdot \sin(\Delta\phi(\mathbf{b})) dx dy \end{aligned}$$

we show together with eq. (6) that the OFI signal is approximately proportional to the refractive index by the relation: $\sin(\Delta\phi(\mathbf{b})) \approx (\Delta\phi(\mathbf{b})) \approx \delta n(\mathbf{b})$. All these principles are related to and are known from back focal plane tracking.

In summary, I still think that this work is in a too speculative stage and the authors cannot convincingly argue that it will make any impact in microscopy as it stands.

Reply: Theory papers show concepts or do predictions and usually use some approximations. I think in this way you can use your speculation argument against any theory paper. However, in our concept paper we showed several results both in theory and experiments.

Reviewer #3 (Remarks to the Author):

The revised manuscript is well improved with some new simulations and explanations. Also, the authors claim the differences between their work and that from Kim & Park. Indeed, this technique could be used to trap larger objects with multiple highly focused point traps. It is obvious, because the focused laser spots have stronger gradient forces. Actually, the two methods share the same roots. Unfortunately, we don't see much new physics from this point.

Reply: I'm not sure what you mean with "Share the same roots". At least both methods try to consider the refractive index of the object to be trapped to enhance the optical forces.

And I don't know what you mean with new physics? If you mean unexpected findings related to physics I would say that OFI tracking or imaging - which precisely locates refractive index changes in scattering media - is unexpected physics! And I would say the concept shown in figure 2 about non-blind optical trapping is rather new as well!

I agree with Reviewer #1 that this work is interesting and useful, but it might be better to publish in a more technical journal instead of Nature Communications.

REVIEWERS' COMMENTS

Reviewer #1 (Remarks to the Author):

I am happy with the changes made to the manuscript in the most recent round of review, and suggest it can be published as it is.